# Hollow-core optical fibre sensors for operando Raman spectroscopy investigation of Li-ion battery liquid electrolytes

Ermanno Miele[1,2,3], Wesley M. Dose[2,3,4], Ilya Manyakin[1], Michael H. Frosz [5], Zachary Ruff[2,3], Michael F. L. De Volder [3,4], Clare P. Grey [2,3✉], Jeremy J. Baumberg [1,3✉] & Tijmen G. Euser [1,3✉]

Improved analytical tools are urgently required to identify degradation and failure mechanisms in Li-ion batteries. However, understanding and ultimately avoiding these detrimental mechanisms requires continuous tracking of complex electrochemical processes in different battery components. Here, we report an operando spectroscopy method that enables monitoring the chemistry of a carbonate-based liquid electrolyte during electrochemical cycling in Li-ion batteries with a graphite anode and a $LiNi_{0.8}Mn_{0.1}Co_{0.1}O_2$ cathode. By embedding a hollow-core optical fibre probe inside a lab-scale pouch cell, we demonstrate the effective evolution of the liquid electrolyte species by background-free Raman spectroscopy. The analysis of the spectroscopy measurements reveals changes in the ratio of carbonate solvents and electrolyte additives as a function of the cell voltage and show the potential to track the lithium-ion solvation dynamics. The proposed operando methodology contributes to understanding better the current Li-ion battery limitations and paves the way for studies of the degradation mechanisms in different electrochemical energy storage systems.

[1] Nanophotonics Centre, Department of Physics, Cavendish Laboratory, University of Cambridge, CB3 0HE Cambridge, United Kingdom. [2] Department of Chemistry, University of Cambridge, Lensfield Road, CB2 1EW Cambridge, UK. [3] The Faraday Institution, Quad One, Harwell Science and Innovation Campus, Didcot, OX11 0RA Oxford, UK. [4] Institute for Manufacturing, Department of Engineering, University of Cambridge, 17 Charles Babbage Road, CB3 0FS Cambridge, UK. [5] Max Planck Institute for the Science of Light, Staudtstr. 2, 91058 Erlangen, Germany. ✉email: cpg27@cam.ac.uk; jjb12@cam.ac.uk; te287@cam.ac.uk

The development of longer-lasting batteries requires a better understanding of the degradation mechanisms that cause Li-ion battery (LIB) failures. For example, the performance loss is particularly critical in batteries with next-generation high energy cathodes, such as Ni-rich $LiNi_xMn_yCo_{(1-x-y)}O_2$ (NMC, x > 0.6)[1], which are designed to last for up to twenty years in, for example, electric vehicle applications[2]. A wide range of damaging mechanisms involving different cell materials and coupled reaction processes have been proposed for these materials[3–12]. Suggested mechanisms include inter- and intragranular cracking of cathodes[3,4] leading to capacity loss and dissolution, and release of transition metal ions, affecting the formation and stability of the anode Solid Electrolyte Interphase (SEI) layer that is essential for battery operation[5,13–16]. Additional destructive processes include lattice oxygen release at the electrode-electrolyte interface (EEI) of the NMC cathode[11]. The onset potential for this is lower for NMCs with higher Ni content[6,7], where oxygen loss results in a structural transformation of the surface[8,17,18]. Oxygen release has also been linked to (electro)chemical degradation of the $Li^+$-solvated carbonate solvents that comprise the electrolyte, such as ethylene carbonate (EC)[6,19–21]. Additives such as vinylene carbonate (VC) are used to improve SEI stability, but their role in cathode processes is less clear[9,10,22]. A particularly sensitive probe of such coupled mechanisms is the electrolyte because it contains a range of reaction by-products, the study of which should thus lead to an improved understanding of the complex interactions that cause the overall cell degradation.

Unfortunately, there is currently a lack of experimental techniques for *operando* monitoring the electrolyte composition and structure within full cells. The ideal sensor would (i) not perturb the device operation, (ii) be nondestructive, (iii) not compromise safety, (iv) work during battery operation, and (v) be easily combined with other cell components without affecting the operation and lifetime of the battery[23]. Suitable sensing probes include the molecular vibrational spectroscopies, FT-IR and Raman[24–26], which have recently gained popularity in battery science due to their ability to provide a safe, fast, and label-free analysis of the chemical composition and evolution of different battery components. Recent examples are the in situ FT-IR analysis of the oxidation of EC at the $LiNi_{0.8}Mn_{0.1}Co_{0.1}O_2$ (NMC811) interface[10], and in situ Raman studies on graphite electrodes[27,28].

The above experiments use configurations with optical access windows at the back of the electrode rather than at the side that faces the separator and investigate processes at either the negative or positive EEI. While half cells are often studied (i.e., the counter electrode is lithium metal), the major drawback of this is that they result in different (electro)chemical behaviour to full cells, and the degradation mechanisms on the separator side may be very different. In addition, their electrode-surface to electrolyte-volume ratio is often much smaller than in commercial cells. As a result, electrolyte measurements in such systems do not capture the complex cross-talk and interplay between cathode, anode, and electrolyte processes within batteries under actual working conditions[29], or allow commercial cells to be studied. Therefore, there is an urgent need to develop non-perturbing optical detection methods for full cells. These would enable studies of the electrolyte's role in more complex degradation processes and failure mechanisms, ultimately leading to real-time electrolyte monitoring in commercial cells.

Fibre-optic devices have emerged as useful nonperturbative sensors within batteries due to their small footprint, non-conductance, and chemical stability. For example, fibre Bragg sensors are used to monitor volume and temperature changes in batteries during cycling[30,31]. More recently, carefully processed silica fibre probes[32] were used to monitor electrolyte composition near the tip of a fibre by Raman spectroscopy[33]. This work was based on conventional all-glass optical fibres, which face several limitations including a relatively weak Raman signal from the short interaction length (limited to the fibre tip) and the silica background due to the solid fibre core, limiting the detection sensitivity.

On the other hand, hollow-core optical fibres (HC-fibres) offer an excellent way to remove the silica Raman contribution and enhance the light-matter interaction length. These HC-fibres comprise a central core channel surrounded by a glass microstructure designed to support low-loss light guidance within a specific wavelength range. Early designs relied on photonic bandgap and interference effects in periodic cladding structures[34]. HC-fibres have since found application in gas-based non-linear optics[35] and as liquid-filled optofluidic microreactors for a range of photochemical and catalytic reactions[36]. In Raman spectroscopy, HC-fibres have been exploited for the flexible and background-free delivery of pump light to a sample[37–39], and silica-background-free Raman sensing in aqueous samples in a hollow-core photonic bandgap fibre[40]. Recently, it has been shown that simplified HC-fibre structures, consisting of a single ring of capillaries surrounding a central waveguide core (see Supplementary Figs. 1 and 2), can also provide excellent low-loss guidance[41–45]. A significant advantage of these fibres is their simplified fabrication process and less complex cladding structure that facilitates the infiltration of liquid samples.

Here we use simplified hollow-core fibre probes to enable operando spectroscopy on full cells, thus overcoming the challenges and limitations facing many cells designed for optical measurements. We repeatedly extract a small electrolyte volume from a working NMC811-graphite LIB into the sub-µL fibre core and analyse its chemical composition by silica-background-free Raman spectroscopy. Finally, we infuse the sample back into the battery and repeat the measurement continuously, resulting in a real-time trace of the electrolyte chemistry during battery cycling.

Multiple studies have found that the solvent EC and additive VC are involved in crucial reaction pathways at the surface of the SEI at graphite anodes and in the bulk electrolyte[10,21,22,46]. The key aim of this proof-of-principle operando Raman study is therefore to directly monitor a broad range of electrolyte chemistries during the formation of the SEI layer during the first electrochemical cycle (formation cycle) of an NMC-graphite full cell[47,48].

## Results

**Hollow-core fibre Raman sensor**. In the Raman setup (Fig. 1a), the proximate end of a 10–15 cm long hollow-core fibre is enclosed in a custom-made ultralow-dead-volume microfluidic cell that allows optical and fluidic access to the fibre (Fig. 1b, c). The distal end of the fibre is fitted and sealed in between the electrodes of a pouch cell. Two layers of monolayer PE polymer separator (MTI) were used to avoid direct contact between the fibre and the electrodes (Fig. 1d). The simplified hollow-core fibre (Fig. 1c) is optimised to guide light in the wavelength range of the Raman pump light and signal[41–43,49] when filled with electrolyte (see Methods and Supplementary Figs. 1–5). The fibre's 36 µm wide core region acts as both a waveguide and microfluidic channel, with a low internal volume of 30 nL per cm fibre length[36]. An automated syringe pump is used to sample and infuse electrolyte from the pouch cell on demand.

A Raman pump laser (785 nm continuous wave, Fig. 1a) is launched into a waveguide mode of the electrolyte-filled fibre core using an underfilled 10x 0.3 NA microscope objective. Raman signals are generated along the length of the fibre, and a portion is captured in backwards-propagating fibre modes and guided back to the proximate fibre facet. The CCD (Charge Coupled Device)

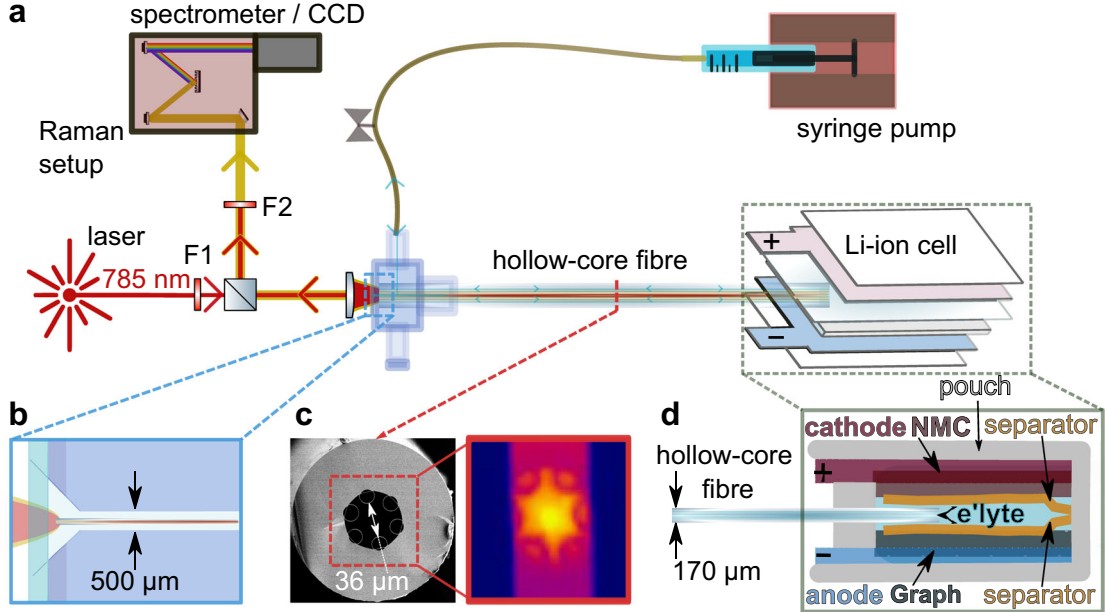

**Fig. 1 Hollow-core fibre-coupled Raman analysis of a Li-ion pouch cell. a, b** Continuous- wave laser light (785 nm) is filtered and launched into the core of a single-ring HC-fibre. The proximate fibre facet is embedded in a microfluidic chamber with a sapphire window (**b**) and connected to a syringe pump for on-demand sampling/infusion. The distal fibre facet is embedded in the pouch cell under study. The backscattered Raman signal generated within the HC-fibre reflects off a 50:50 beam-splitter, filtered and analysed by a spectrometer. F1, bandpass filter; F2, notch filter. **c** SEM image of the hollow-core fibre with an outer diameter of 174 μm and a core diameter of 36 μm (measured between the inner capillaries (see Supplementary Fig. 1). The right-hand image shows the Raman signal detected by the spectrometer CCD (Charge Coupled Device) camera. **d** Schematic indicating the electrodes, separator, and fibre probe arrangements within the pouch cell (not to scale—for details, see Supplementary Figs. 3 and 4).

image of the generated Raman light (right-hand image in Fig. 1c) shows that most of the Raman light is generated and guided within the hollow fibre core. After each optical measurement, the electrolyte sample is injected back into the pouch cell. Avoiding any cell perturbations leads to single sampling intervals that take 22 min (ca. 4% of the full discharging time at C/10 C-rate). Sampling was repeated periodically to monitor the electrolyte over extended periods (a typical charge-discharge formation cycle takes more than 10 h).

**On-line Raman collection in a hollow-core fibre.** The dynamic exchange and Raman spectroscopy within the fibre core were first tested ex situ without the pouch cell for a range of electrolyte components and typical solvents (Fig. 2). The spectrometer CCD records proximate facet images and spectrally-dispersed fibre images (Fig. 2a). Spectra were continuously recorded with a 20 s integration time per spectrum throughout the experiment. To enable simultaneous monitoring of multiple Raman bands, we made an optimal trade-off between spectral range, resolution, and signal strength (Fig. 2b); see also Methods section. For experiments in Figs. 2–5, a coarse diffraction grating was used to capture a broadband Raman signal. Later, as a proof of principle, ex situ high-resolution Raman spectra (over a smaller spectral window) were captured using a higher-density grating (Supplementary Figs. 6a–c).

Initially, the fibre was filled with isopropyl alcohol (IPA), whose Raman spectrum is shown in Fig. 2b–c. Syringes were swapped to exchange samples, and the pump flow rate was set to 5 μL/min (0.083 μL/s) to infiltrate the fibre core. The syringe pump is switched off once the Raman signal is stabilised. The fluidic stabilisation time of the system after sample exchange is currently ~400 s (corresponding to a flow volume of ~33 μL, Fig. 2c). Samples sequentially infiltrated into the fibre here are IPA, ethyl methyl carbonate (EMC), a 3:7 mixture of ethylene carbonate

(EC) and EMC, and the commercial battery-grade liquid electrolyte solution LP57 (i.e., 1.0 M LiPF in EC:EMC 3:7 v/v). For each sample, relatively broadband Raman spectra were taken (Fig. 2c) between 410 and 2182 cm$^{-1}$.

The Raman spectra show clear signatures of the various electrolyte components. First, the spectral position of the PF$_6^-$ anion Raman band at 740 cm$^{-1}$ can clearly be seen in the LP57 electrolyte[50]. The PF$_6^-$ peak partially overlaps with the EC-skeletal mode at ~720 cm$^{-1}$, as can be seen in our high-resolution scan of this band (Supplementary Fig. 6b)[51]. The ability to detect PF$_6^-$ is helpful since its decomposition is a proposed degradation mechanism occurring at the surface of nickel-rich cathodes such as NMC811[9]. Furthermore, PF$_6^-$ reacts readily with the water generated in electrolyte decomposition reactions[9]. Secondly, the EC breathing mode at 893 cm$^{-1}$ is connected to the ring structural integrity of the molecule. Finally, the shaded (broad purple) band between 1700−1850 cm$^{-1}$ corresponds to Raman peaks of the carbonyl (C = O) bonds in EMC and EC/VC[52,53], whose spectral positions change with lithium-ion solvation dynamics (see Fig. 4 and Supplementary Fig. 6c)[25,26,51,54–56]. Also marked is the expected position of the (weak) spectral band at 1628 cm$^{-1}$ (dotted grey line) due to vinylene –HC = CH-vibrations of the additive VC (explored below in Fig. 3). Therefore, by using a low-density diffraction grating in our setup, we monitor many of the important electrolyte components simultaneously.

**Operando Raman measurements in full cells.** A hollow-core fibre is embedded in a pouch cell composed of a LiNi$_{0.8}$Co$_{0.1}$Mn$_{0.1}$O$_2$ (NMC811) cathode and a graphite anode to monitor its electrolyte chemistry operando, i.e., during cycling. Each circular electrode has an active area of 1.54 cm$^2$ (14 mm diameter) and is covered by a layer of polymer separator (see Methods). The HC-fibre is placed between the two separator layers to protect the electrode surfaces from

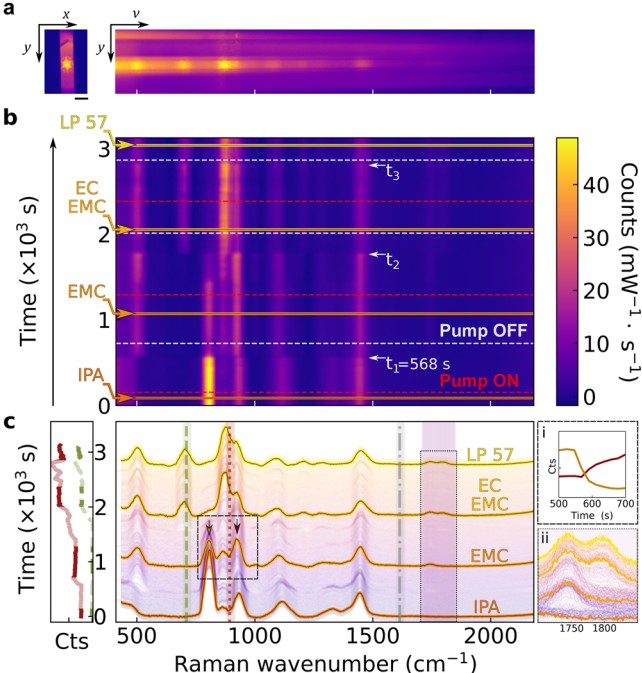

**Fig. 2 Online Raman measurements in a hollow-core fibre. a**, Raman light emerging from the fibre facet (left, image scale bar is 50 μm) with a spectrally dispersed image (right). **b**, Raman spectra tracked during sequential sample infiltration. Red dashed lines indicate when the pump switched on; $t_{1-a}$ indicate the times at which the sample fluid reaches the fibre core. Dashed white lines indicate when the pump is switched off, followed by a switch of the sample syringe. Solid horizontal lines indicate times at which the spectra shown in **c** were taken. **c**, obtained spectra of the different solvent mixtures. Highlighted Raman bands relevant to battery chemistry are ethylene carbonate breathing mode at 893 cm$^{-1}$ (dark red dotted line), PF$_6$ anion mode at 740 cm$^{-1}$ (green dashed line), and vinylene carbonate -HC = CH- band centred at 1628 cm$^{-1}$ (not present in these solutions and spectra). The shaded area indicates EC and EMC bands at 1700−1850 cm$^{-1}$ relevant to lithium solvation mechanisms (see Fig. 4) Inset **i** demonstrates the sample exchange time as monitored by the Raman intensity of the IPA (819 cm$^{-1}$) and EMC skeleton (~900 cm$^{-1}$) modes (arrows in **c**). Inset **ii** shows rescaled EC and EMC bands at 1700−1850 cm$^{-1}$.

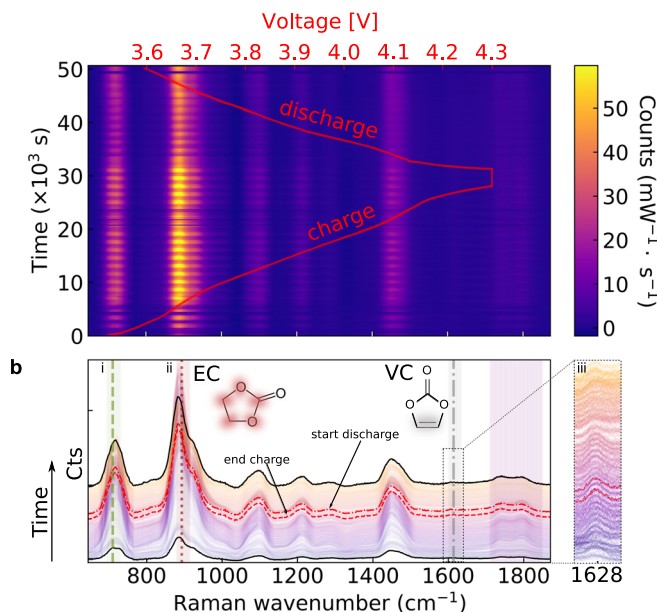

**Fig. 3 Battery electrolyte Raman spectral evolution during cycling. a**, Operando Raman spectroscopy performed during the formation cycle for a LiNi$_{0.8}$Co$_{0.1}$Mn$_{0.1}$O$_2$ (NMC811) - graphite Li-ion pouch cell with an electrolyte comprising LP57 + 2 wt.% VC. The cell was charged galvanostatically to 4.3 V, potentiostatically held at 4.3 V, and then discharged. **b**, Raman spectral evolution showing a range of Raman modes of battery electrolytes, particularly: (**i**) PF$_6^-$ anion, symmetric stretch (740 cm$^{-1}$, green dashed line), (**ii**) EC, skeletal breathing mode (893 cm$^{-1}$, dotted red line), (**iii**) VC, -HC = CH- (1628 cm$^{-1}$, grey dash-dotted line, displayed in inset where the signal was multiplied by 4, relative to that shown in (**b**)).

The evolution of the Raman spectra is measured as a function of the cell voltage (red curve) during the first electrochemical cycle, during which many chemical changes due to EEI formation are expected (Fig. 3a). Clear signatures are observed in the spectral lines for PF$_6^-$, the EC breathing mode, and the carbonyl (C = O) bonds in EMC and EC/VC, as identified in Fig. 2b. In addition, a (weak) vinylene -HC = CH- Raman band at ~1628 cm$^{-1}$ is detected and will be discussed more detail below. Collecting the full Raman spectra throughout the cycle allows for a detailed analysis of the electrolyte salts and solvents and their interactions.

**Tracking lithium solvation dynamics**. How lithium ions are solvated by the carbonate solvent has important implications for the interfacial reactions at both the graphite and layered metal oxide EEI[58–60]. In Raman, the band between 1700−1850 cm$^{-1}$ is related to vibrations of the C = O group, whose oxygen atoms are primarily responsible for solvating the lithium ions[26,61–63]. The Raman bands in the spectral region observed at 1700−1850 cm$^{-1}$ (Fig. 4) illustrate the evolution of cyclic and linear carbonate C = O stretches during the formation cycle (Fig. 4b). The overall intensity of the EC C = O band during the first cycle follows a similar trend (increase followed by a decrease) to the EC breathing mode in Fig. 5.

Lithium salt concentration in the electrolyte and the interfacial chemistries at the EEIs can alter the solvation structure of carbonates solvents[10,25,55,56,64]. To further calibrate our methodology, ex situ test measurements were carried out by directly infiltrating EC:EMC mixtures with different Li concentrations into a hollow-core fibre sensor. The resulting data shows that the EC breathing-mode intensity doubles when changing the Li concentration from 0.5 to 1 M (Supplementary Fig. 7). In addition, the

mechanical damage by the fibre (Fig. 1d). The cell is sealed and filled with 100 μL of LP57 with 2 wt.% VC added. Details of the cell components and their assembly are in Supplementary Figs. 3 and 4. Even though the HC-fibre creates a slight spacing between the two separators, the total electrode surface to electrolyte volume ratio (~15 cm$^2$/mL) remains very close to that of pouch cells assembled routinely in research environments[57].

The cell is galvanostatically charged to 4.3 V, potentiostatic held for 1 h at 4.3 V, and finally discharged to 3.5 V, at a cycling rate of C/10 (18.5 mA g$^{-1}$$_{NMC}$). To ensure complete sample exchange in the fibre-core, a 24 μL volume micro-sample (ca. 50 times the internal fibre volume) is extracted from the battery every 22 min, analysed by in-fibre Raman spectroscopy, and re-injected into the pouch cell. We estimate the time resolution required to monitor electrode processes from the diffusion time $t_d$ of EC molecules across the separator (from the cathode to the anode)[46]. Assuming a polymer separator tortuosity of 2.5 and a liquid diffusion coefficient of 10$^{-6}$ cm$^2$/s[46], this results in a diffusion time for molecules to travel from one electrode the other of $t_d = 445$ s (~7 min) for our cell geometry. As in the previous experiment, we used a broad spectral window (640−2340 cm$^{-1}$, coarse grating) to simultaneously track a range of chemical species.

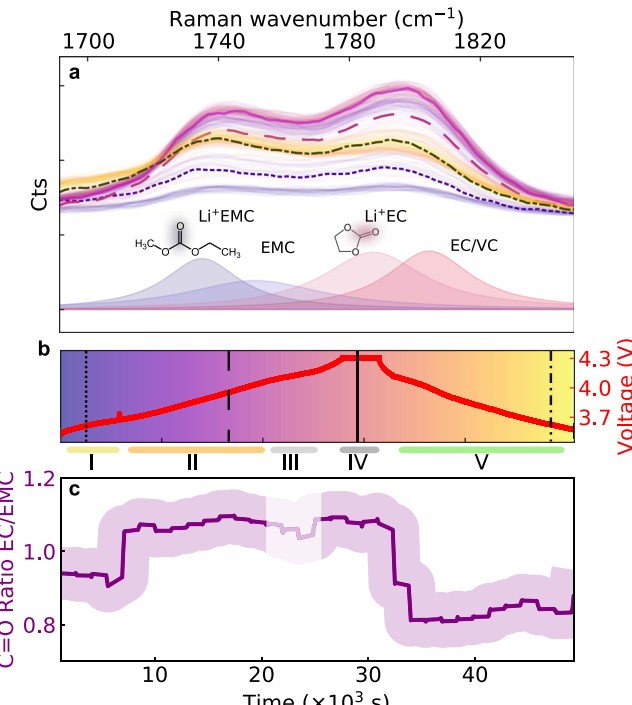

**Fig. 4 Monitoring Li$^+$ solvation to the different solvent carbonyl groups. a, b** Evolution of the carbonyl group (C = O) Raman bands during the formation cycle. Dashed and solid curves are spectra taken at times highlighted in **b** The deconvolution based on the solid purple line in **a** suggests that the peaks comprise a combination of solvent C = O stretch bands for un-solvated Li$^+$ (dark blue, ~1750 cm$^{-1}$, for EMC and dark red, ~1805 cm$^{-1}$, for EC) and solvated Li$^+$ (light blue, 1735 cm$^{-1}$, for Li$^+$EMC and dark red, ~1787 cm$^{-1}$, for Li$^+$EC). With the broadband (and thus lower-resolution) grating used here, differences between solvated and non-solvated Li$^+$ cannot be simply resolved. **c,** Trend of C = O band intensity ratio between EC C = O (solvated and un-solvated 1782–1817 cm$^{-1}$) and EMC C = O (solvated and un-solvated bands, 1730–1765 cm$^{-1}$). The shaded region indicates a 10% confidence interval (estimated from typical fluctuations in the Raman signal). The complex interplay of lithium solvation dynamics with solvents (EC, EMC) and additive (VC) is segmented into different regimes (coloured horizontal lines I-V) with a different behaviour at each distinct state of charge.

EC C = O band doubles its strength relative to EMC C = O for the same Li$^+$ concentration change. This calibration suggests that the observed spectral differences of the two bands centred around ~1740 and ~1800 cm$^{-1}$ in Fig. 4a are at least in part related to changes of the lithium solvation at the carbonate solvents. In particular, the evolution of the balance between solvated cyclic carbonate (Li$^+$- O = C, EC) and solvated linear carbonate (Li$^+$- O = C, EMC) can be tracked indirectly from the ratio between the intensities of the EC C = O band (solvated and un-solvated, 1782–1817 cm$^{-1}$) and the EMC C = O band (solvated and un-solvated, 1730–1765 cm$^{-1}$) during a cycle (Fig. 4c).

To help analyse the observed dynamics, we divide the electrochemical cycle into five different regimes. Initially (up to 3.75 V – region I), the EC C = O and EMC C = O bands maintain a constant and similar intensity. Then, as the cell voltage increases, both Raman bands increase in intensity (region II), with the rise for EC C = O being significantly higher. Finally, at around 4.1 V (transition IV-V), during discharge, there is a sudden decrease in both the EC C = O and EMC C = O bands, with the largest reduction observed for EC C = O.

**Monitoring ethylene carbonate and vinylene carbonate dynamics.** It is well-understood that both EC and VC are reduced during the initial cycle to form the SEI[14]. Therefore, we monitor the EC breathing mode (Fig. 5a) and vinylene -HC = CH- band (Fig. 5b) during the formation cycle. The EC intensity changes significantly during different stages of the cycle (Fig. 5c): First, at 3.75 V (t = 6,000 s), the EC band intensity rapidly increases by a factor five (from ~500 to ~2400 counts per sec near transition I-II). It then stays constant until around 4.1 V (t = 20,000 s). Subsequently, the optical signal is temporarily lost, likely due to gas evolution (shaded region III in c). We indeed observe bubbles flowing from the battery into the hollow-core fibre via their side-scattered light (see Supplementary Movie 1). During the voltage hold, the signal returns (no bubbles are detected). Finally, during discharge at around 4.1 V, the EC intensity decreases sharply again (transition IV-V). On the other hand, the vinylene band increases very slightly during the formation cycle (grey curve in Fig. 5d), although we note the relatively low signal-to-noise ratio of this peak. This result is somewhat surprising because, while some VC generation as a result of electrolyte oxidation has been proposed[12,21], consumption of VC to form the SEI during the formation cycle is also expected. This result is discussed further below.

**Multi-cycle measurements.** Operando experiments over multiple charge/discharge cycles were carried out to study changes in the electrolyte chemical composition over extended periods. To investigate whether vinylene species are generated within the cell, the pouch cell was filled with LP57 electrolyte without the VC additive. A silica glass capillary (150 μm outer diameter, 75 μm inner diameter) was inserted between the graphite and square NMC811 electrodes with a surface area of 4 cm$^2$ (see Supplementary Figs. 3 and 8). The electrolyte volume was increased to 260 μL, to achieve an electrode surface area to electrolyte volume ratio of ~16 cm$^2$/mL, similar to the value of ~15 cm$^2$/mL used in the operando measurements in Figs. 3–5. During Raman measurements, the capillary was connected to a HC-fibre using low-dead-volume microfluidic fittings. We note that this approach allows multiple cells to be analysed using a single microfluidic pump and Raman apparatus, aiding long-term multiple-cell cycling studies in the future.

Before measuring any Raman spectra, the cell is electrochemically charged and discharged, and voltage profile graphs are taken during multiple cycles (Fig. 6a). The cell retains ca. 90% of its discharge capacity after 13 cycles (compared to the first-cycle discharge). This capacity loss is comparable to the capacity lost during formation as reported for similar cells NMC811 cells[65], supporting that the insertion of the glass micro-capillary does not substantially affect the electrochemical performance of the cell. During cycle 7, the cell was connected to the HC-fibre, and an operando Raman measurement was performed (Fig. 6b, top shows the charge and discharge voltage profile). Interestingly, the EC breathing mode remains stable throughout the cycle (Fig. 6b bottom, Raman bands intensity), in stark contrast with the large fluctuations observed during formation-cycle measurements (Fig. 5). Furthermore, no bubble formation is observed during this seventh cycle. Finally, a new peak is seen at the position of the vinylene C = C stretch mode, and the formation of vinylene species by electrolyte oxidation becomes more apparent when comparing the spectrum at the end of cycle 7 to the fresh electrolyte (Fig. 6c). The data confirm that the large EC fluctuations, the increases in vinylene species, and bubbles (Fig. 5) are indeed related to electrochemical processes during the first cycle[66].

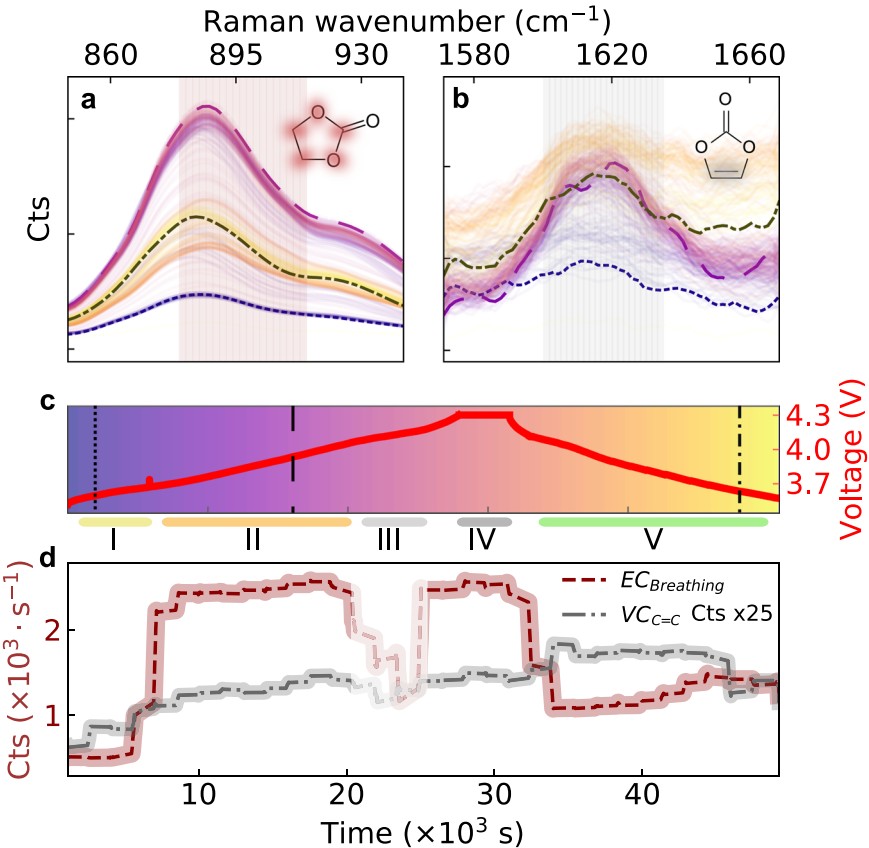

**Fig. 5 Monitoring Ethylene Carbonate (EC) solvent and vinylene concentrations. a**, **b** Evolution of Raman EC-breathing mode and vinylene -HC = CH- band during the formation cycle (baseline-subtracted). Dashed curves are spectra taken at times highlighted in **c** (colouration of each curve corresponds to a specific time). Cell voltage during cycling, segmented into regions I-V. **d** Time evolution of EC and VC Raman band intensities. The Raman signal intensity of the vinylene band in **b** was multiplied by 25 relative to that shown in **a**.

## Discussion

This *operando* spectroscopy gives access to many Raman signatures not previously accessible in full cells. We thus provide tentative explanations to highlight which processes can be studied using such HC fibre-based operational techniques.

**Li⁺ solvation**. The increase of EC C = O relative to the EMC C = O Raman mode measured in regions II-IV (Fig. 4c) could be related to Li⁺ concentration changes and/or Li⁺ solvation of the solvent molecules[67], which is consistent with the ex situ data in Supplementary Fig. 7. However, the increase could also be caused by changes in the bulk EC/EMC ratio, which cannot be resolved at the spectral resolution used in Fig. 4. The EC/EMC C = O band ratio (Fig. 4c) reduces after the voltage hold at maximum charge. This phenomenon requires further study but may be related to different concentration gradients in the cell during charging/discharging. We note that related increases in carbonate IR absorption bands at ~1800 cm⁻¹ were observed in surface-sensitive in situ FT-IR data in NMC811 half cells in Ref. [10], which attributed this to electrolyte oxidation at the Ni-rich electrode surface. Our operando Raman data, on the other hand, detects these increases away from the surface, in the bulk electrolyte. The Li⁺ solvation structure in EC and EMC may thus also affect the Li⁺ diffusion coefficient in the electrolyte[61] with implications for battery operation.

**Vinylene-concentration changes**. We measure a slight increase (Fig. 5d) of the Raman band -HC = CH- associated with vinylene species (-HC = CH-) in contrast with the predicted consumption of VC at the anode during the formation cycle and therefore a

reduction of the -HC = CH- band intensity. A similar vinylene band is observed after 7 cycles in a pouch cell assembled with VC-free electrolyte (Fig. 6c). To identify what vinylene species are generated, we carried out combined Raman and ex situ ATR FT-IR studies of electrolyte extracted from a coin cell cycled 10 times initially filled with VC-free LP57 electrolyte (see Supplementary Fig. 9). This experiment reproduced the appearance of a vinylene Raman band at ~1628 cm⁻¹. Since this peak is not present in the FT-IR spectrum of the VC-containing electrolyte (Supplementary Fig. 9c) this suggests that this band is related to linear vinylene species rather than the symmetric cyclic VC molecule. VC may be formed at the cathode via oxidation or dehydration of EC[9,10], accounting for the small increase in VC peak intensity after the potentiostatic hold (Fig. 5d), which is one proposed mechanism for VC formation involving lattice oxygen release from NMC in the form of reactive singlet oxygen[21]. However, a signal from a C = C stretch is also seen in the pouch cell after 7 cycles without VC in the starting electrolyte (Fig. 6c) and in the ex situ data from the 10 times cycled coin cell (Supplementary Fig. 9c). Raman and FTIR spectra of an aged electrolyte show that this cannot be assigned to VC but rather to an asymmetric linear vinylene-containing molecule such as observed in FT-IR studies on chemically reduced electrolyte by Ref. [68]. These linear C = C species are also likely to result from VC, but via further reduction reactions at the anode, motivating further joint Raman/IR studies of VC reduction.

**Bubble formation**. The temporary loss of optical signal during the final stage of the charging cycle is confirmed to be due to the

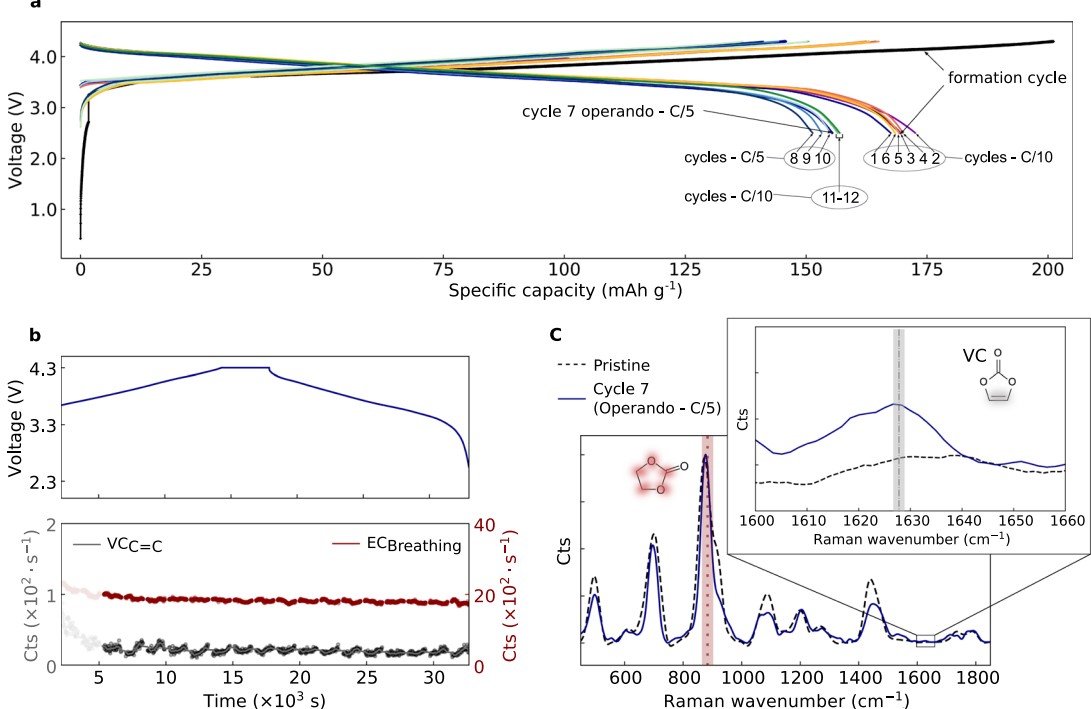

**Fig. 6 Long-term cycling and Raman analysis. a** Charge/Discharge voltage profiles of a Li-ion pouch cell (anode: graphite, cathode: NMC811, electrolyte: LP57 without VC additive) cycled at C/10 and C/5 rates. Labels 1–13 correspond to the number of each discharge cycle. **b** Operando analysis and time evolution of EC (red dashed line) and vinylene (grey dash-dotted line) Raman band intensities during cycle 7. Note that only Raman data for $t > 5 \times 10^3$ s are shown, by which time the fluid sampling has stabilised after reconnecting the cell to the HC-fibre. The small periodic modulation in intensity is related to the fluidic sampling interval. **c** Comparison between Raman spectra of pristine (not cycled) LP57 electrolyte (dashed line) and electrolyte extracted at the end of cycle 7. The inset in **c** shows the vinylene Raman band detected in the cycled electrolyte.

formation of bubbles. Since they are observed in several formation-cycle experiments (see Supplementary Fig. 10a–b), but are largely absent during later cycles (Fig. 6), these are likely related to gas generation in EEI formation reactions during the formation cycle (CO, $CO_2$, $C_2H_4$)[69]. In the current configuration, detection of such bubbles within the central compartment is delayed, as gas need to diffuse through the separator and coalesce into bubbles large enough to be detected in the HC-fibre. We note that, in principle, our capillary setup provides a straightforward method to couple the pouch cell with a mass spectrometer for gas analysis[69].

**EC-Raman band**. The EC breathing band between 880–915 cm$^{-1}$ shows a significant evolution in intensity during the formation cycle (Fig. 5d), consistent with the trend in the ratio of EC/EMC C = O stretch (Fig. 4c). During the first cycle, we observe an initial five-fold increase in the EC breathing band. However, we note that the operando Raman data in cycle 7 (Fig. 6b) do not show significant EC-band fluctuations, supporting the hypothesis that these changes are related to the formation cycle. To check how the first-cycle increase in EC-signal is affected by variations in Li$^+$ concentration, calibration experiments were performed in which the Li$^+$ concentration was varied from 0.5 to 1 M, resulting in an at most two-fold increase in Raman intensity (Supplementary Fig. 7). The EC breathing band at 880–915 cm$^{-1}$ comprises two distinct modes at 893 cm$^{-1}$ (not solvated) and 903 cm$^{-1}$ (solvated), which are resolved in the high-resolution spectra in Supplementary Fig. 6. The intensity ratio between these two modes is known to be particularly sensitive to Li-ion coordination[25,64].

It is also consistent with earlier FT-IR studies in a half-cell configuration[10] that showed that the peak position and intensity of the EC C = O IR absorption band is affected by the Li$^+$ concentration and that EC undergoes a progressive

dehydrogenation correlated with the cathodic potential[10]. Furthermore, changes in Li$^+$ solvation will increase the refractive index of the electrolyte by typically 0.001 per Li$^+$ molar concentration, as was demonstrated with different salts and salt concentrations[70]. This would shift the transmission window of the fibre by up to ~4 nm (Supplementary Fig. 5b), and slightly affect the amount of Raman pump light propagating through the hollow-core fibre[41]. However, even when taking into account all the above effects, the five-fold increase in EC-Raman signal cannot be fully explained by changes in Li$^+$ concentration alone, suggesting that other unknown effects may play a role.

To conclude, hollow-core fibre sensors were embedded in working Li-ion batteries to measure silica background-free Raman spectra of electrolytes during electrochemical cycling. These integrated sensors identified changes in the electrolyte of a Li-ion battery comprising a commercially relevant high energy Ni-rich layered oxide cathode (NMC811) and a graphite anode. As a proof-of-principle, broadband electrolyte Raman spectra were continuously acquired during the formation cycle, during which (among other things) the solid electrode interphase (SEI) is created. In addition to the broadband operando study, ex situ high-resolution Raman spectra were measured to demonstrate the ability to perform targeted investigations into specific chemical processes within Li-ion cells.

We have observed changes in the Raman spectrum related to the C = O stretch modes of carbonate solvents during cycling and have measured changes in the concentrations of vinylene- (C = C) double bonds. Multi-cycle measurements without the initial VC additive confirm that the fluctuations in the EC signals are related to the formation cycle. Finally, ex situ ATR FT-IR studies show that the vinylene Raman signal that appears after the initial cycle originates from a linear vinylene species rather than cyclic VC.

Our results show the potential of hollow-core fibre spectroscopy to study how (electro)chemical degradation of electrolytes affects lithium solvation. Future work employing the narrower-band, high-resolution spectrometer configuration, will clearly track the solvated and non-solvated Raman modes of various targeted electrolyte components. Operando electrolyte monitoring can facilitate the study of complex chemical pathways and cross-talk between chemical species in a real-world battery. A key example is the appearance of a vinylene Raman band in a sample without initial VC, suggesting that, while vinylene species are consumed at the anode[21,66], they are also generated through EC oxidation during cycling, as was proposed by Refs. [12] and [21]. The operando broadband spectra allow simultaneous tracking of a wide range of Raman bands of different electrolyte components, while the multi-cycle experiments show the potential to study interactions between novel electrolytes and electrodes during repeated cycling.

## Methods

**Battery assembly and cycling**. Full cells were assembled into a pouch cell in a dry room with a dew point less than $-53\,°C$ (<150 ppm moisture) and an area of ~18 m². Pouch components (foil, current collector tabs, PE polymer separators, hot melt adhesive tape and strapping tape) were purchased from MTI (Pi-KEM, Tamworth, UK). Graphite anodes and $LiNi_{0.8}Co_{0.1}Mn_{0.1}O_2$ (NMC811) cathode sheets were kindly provided by the Argonne National Laboratory Cell Analysis, Modelling, and Prototyping (CAMP) facility. The electrodes were single-side coated in a climate-controlled dry room with a dew point less than $-42\,°C$ (<100 ppm moisture) at a rate of 0.4 m min⁻¹ using a pilot-scale semi-automatic coater (A-PRO Co.). Graphite electrodes consisted of 91.83 wt.% Hitachi MagE3 artificial graphite, 2 wt.% Timcal C45 conductive carbon, 6 wt.% Kureha 9300 polyvinylidene difluoride (PVDF) binder, and 0.17 wt.% oxalic acid printed onto 10 µm thick copper foil. The Cu foil is planar and has a purity of 99.9 %. The average thickness of the negative electrodes was 42 microns (excluding the Cu foil thickness). Electrodes were calendared at 0.5 m min⁻¹ using a heated (80 °C) two-roller hydraulic-driven roll press (A-PRO Co.) to 30 % porosity. NMC811 electrodes were made up of 90 wt.% Targray NMC 811, 5 wt.% Solvay 5130 PVDF binder, and 5 wt.% Timcal C45 conductive carbon on 20 µm thick aluminium foil. The aluminium foil is not etched and has a purity of 99.385 %. The average thickness of the positive electrodes (NMC) was 33 microns (excluding the Al foil thickness). Pre-cut electrodes (round, 14 mm dia. NMC cathode and 15 mm dia. graphite anode for the experiments in Figs. 3–5, and 20x20 mm square electrodes for experiments in Fig. 6) were dried in a vacuum oven (120 °C for >12 h) and stored in a glove box (<0.5 ppm $H_2O$ and $O_2$) before use. The cathode and anode electrodes had active mass loadings of 8.21 $mg_{NMC}\,cm^{-2}$ (corresponding to ~1.52 mAh cm⁻² or 185 mAh g⁻¹_NMC) and 5.83 $mg_{Gr}\,cm^{-2}$ (~1.92 mAh cm⁻² based on 330 mAh g⁻¹_Gr), respectively. The n:p ratio (*i.e.*, the fraction of capacity on the anode (n) relative to the cathode (p)) for these electrodes is calculated to be in the range 1.16–1.26. Battery electrolytes, solvents, and additives (LP57, EC:EMC 3:7) were purchased from SoulBrain, VC was purchased from Solvionic. The maximum water content in the solvents was 30.3 ppm (LP57). The hollow-core fibre was placed between electrodes and two polymer separator layers to prevent contact with the electrode surfaces (see Supplementary Fig. 3). Typical dimensions of assembled cells are 6 × 4 cm². Cells were cycled with an Autolab PGSTAT204 (Metrohm) potentiostat at a C-rate of C/10 (based on 185 mAh g⁻¹_NMC) with a constant current (CC) charge to 4.3 V, constant voltage (CV) hold step for 1 h, and CC discharge to 3.5 V. The cells were cycled in a room with regulated temperature (20–22 °C) and humidity (45–55%). For the long-term cycling data in Fig. 6a formation cycle is followed by 6 cycles at C/10, 4 cycles at C/5 and finally 3 cycles at C/10.

For the ex situ FT-IR and Raman studies in Supplementary Fig. 9, a coin cell was assembled from the same electrode materials as described above (NMC 811 cathode, graphite). A 1 × 17 mm dia. glass fibre separator (GF/A, Whatman) was placed between 2 × 16 mm dia. Celgard separators, which are soaked with 140 µL of LP57 electrolyte with no VC added. The cell was cycled 10 times between 2.5–4.2 V at a C/20 rate assuming a practical capacity for NMC811 of 185 mAh/g. At cycle 10, the cell was opened under Ar atmosphere. To extract electrolyte from the coin cell, its glass fibre separator was inserted in a falcon tube and centrifuged for 30 min at 4500 rpm. An Ar-filled glass jar was used as a container to transport electrode- and electrolyte samples from the Ar-filled glovebox to the setup used for the ex situ measurements.

**Hollow-core fibre design**. The fibre used is a simplified hollow-core photonic crystal fibre fabricated using a stack and draw process[34]. The fibre preform was created by positioning six thin-walled silica glass capillaries in the corners of a larger tube with a hexagonal hole. This stack was first drawn into several smaller secondary preforms ("canes"), and then into fibre. Silica glass was obtained from Heraeus (Suprasil 300) and has a refractive index of 1.454 at a 785 nm wavelength. Detailed Scanning Electron Microscopy (SEM) analysis of the hollow-core fibre

and its photonic structures are shown in Supplementary Fig. 1. The outer diameter of the fibre was 174 µm, the six inner capillaries have inner diameters between 16–18 µm. The resulting diameter of the hollow core (measured from capillary to capillary) is 36 µm. The wall thickness of the internal capillaries is between ~410 and 440 nm, with an average value of $t = 425$ nm.

The guidance properties of this type of microstructured fibre are predicted by the anti-resonant reflection (ARROW) model[41]. The ARROW model calculates resonant wavelengths (for which light can resonantly leak out of the fibre) and anti-resonant wavelengths (for which the cladding layer acts as a mirror and light is guided along the fibre). For a fibre filled with a typical electrolyte mixture (refractive index at 785 nm $n_1$ ~1.39) and a glass refractive index of $n_2 = 1.454$, we can predict the first anti-resonant wavelength to be $\lambda_{AR} = 4t\sqrt{n_2^2 - n_1^2} = 725$ nm, close to the wavelength range relevant to the Raman spectroscopy experiments laser (see Supplementary Fig. 5a). The typical transmission of light coupled through an electrolyte-filled fibre is 10–20%. We note that the fibre guidance properties in this type of fibre are relatively robust against changes in electrolyte refractive index (see Supplementary Figs. 2 and 5). However, some fluctuations in transmitted power with core wavelength do appear. The ability to probe a range of electrolyte mixtures with different refractive indices is evidenced by Fig. 2b.

**Microfluidic pressure cell**. We minimised the compliance and dead volume of the fluidic connections to reduce the fluidic response time of the system during sample exchanges. The proximate end of the fibre is connected to a specially designed low-volume stainless-steel cell, covered by a sapphire optical access window (Edmund Optics), resulting in an internal volume of 230 nL. The cell is connected to an automated syringe pump (New Era Pump) actuating a 50 µL Hamilton syringe. The response time of the low-dead-volume microfluidic circuit was estimated to be 1–2 min and is currently limited by the internal volume of the syringe[71]. The geometry of the fibre embedded in the pouch cell is detailed in Supplementary Figs. 3–4 and 8. The microfluidic sampling line was assembled with Polyether ether ketone (PEEK) parts and Zero-Dead-Volume (ZDV) fittings (IDEX Health & Science, LLC, USA).

**Raman collection**. Spontaneous Raman measurements were recorded on a custom-built Python-automated Raman setup. Excitation and collection were through a 10 × 0.3 NA Olympus Plan Fluorite objective. Spectra were recorded by a Pixis 1024-element cooled CCD camera coupled to an Acton SP-2300i Spectrometer (Teledyne Princeton Instruments, USA). A grating with a period of 300 grooves per mm was combined with a 190 µm entrance slit to monitor multiple Raman bands simultaneously. This setting resulted in a broad detection window (644–2344 cm⁻¹) with a resolution of ~27 cm⁻¹. We note that the resolution can be improved to 2.4 cm⁻¹ by using a 1200 lines/mm grating at the cost of a decreased spectral range, as shown in Supplementary Fig. 6. A Python interface code performed automated scans and syringe pump control. For each spectrum, baseline subtraction was performed (details in Supplementary Fig. 11).

**Reporting Summary**. Further information on research design is available in the Nature Research Reporting Summary linked to this article.

## Data availability

Source data are provided with this paper and can be accessed at: https://doi.org/10.17863/CAM.81189.

The instrument control code used in this work can be found on our Nanophotonics GitHub repository https://github.com/nanophotonics. Source data are provided with this paper.

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

## Acknowledgements

This work is supported by the Faraday Institution under grant no. FIRG001 (EM, WMD, ZR, MDV, CPG, JJB, TGE) and the Winton Programme for the Physics of Sustainability (EM, TGE). The authors are grateful to A. Jansen, S. Trask, B.J. Polzin, and A.R. Dunlop at the US Department of Energy's CAMP (Cell Analysis, Modeling, and Prototyping) Facility, Argonne National Laboratory, for producing and supplying the electrodes in this work, and R.P. Mouthaan for help with developing the Python instrument interface.

## Author contributions

T.G.E., J.J.B., C.P.G., and E.M. conceived the concept. E.M., T.G.E., and J.J.B. designed the optical setup, and E.M. conducted the optical experiments, aided by I.M. Battery samples were prepared by E.M. W.M.D. and Z.R. The hollow-core fibres were designed and fabricated by M.H.F. All authors analysed the data and co-wrote the paper.

## Competing interests

EM, JJB, CPG, and TGE are applicants and inventors on a patent on the fibre-optic sensing apparatus and method discussed in the manuscript, patent application number: P/80651.GB01/SG (pending). The remaining authors declare no competing interests.
