## [Peer Review File · Nature Communications]

REVIEWER COMMENTS

Reviewer #1 (Remarks to the Author):

In this paper the authors report a new in situ fiber Raman spectroscopy of batteries. New detailed electrolyte chemistries could be observed. This paper can be published with minor revision.

The authors claim that the spectra obtained by the method are background free. An example of spectrum that clearly shows the background level should be presented (where is zero in Y scales ?).

Reviewer #2 (Remarks to the Author):

The operando method described in the paper is interesting and to my knowledge novel. The method itself, as noted by the authors can most likely bring interesting results, knowledge and thinking to the field of electrolytes and batteries. However, I find that the analysis of the spectra needs to be reevaluated to better validate the claims of the paper. Specific comments are given in the points below.

1. All of the measurements are displayed in units of counts. How was it ensured that the overall intensity of the spectrum did not change over time? Were the spectra normalized? Is the same trend in the various modes seen if the normalized and integrated peak area is used as the evaluation criteria?
2. The Raman resolution of 10 and 40 cm^{-1} is much lower than normally used in electrolyte studies (1-4 cm^{-1}). What implications does this have on the data analysis? How does this impact the separation of Li-coordinated and non-Li-coordinated peaks of the various species?
3. Do the authors expect any effects due to convection as a result of removing and reintroducing the electrolyte in the cell? Do the authors expect any effect due to the reduction of the electrolyte amount by 25% during measurement? What is the electrolyte excess?
4. Page 2, Line 29-30 – is there a reason to believe that degradation of high-energy cathodes is higher in electric vehicle batteries than in batteries for other applications with the same chemistry?
5. Page 4, line 83-84: what challenges exactly are you referring to in this passage? Furthermore does this refer to all spectroscopy methods or only Raman spectroscopy methods.
6. Lines 28-43: The entire first paragraph of the manuscript is dedicated to the discussion of degradation and aging mechanisms. However, in the last paragraph of the introduction, the authors state that the proof of concept cell is used to monitor electrolyte chemistry during the first cycle during SEI formation. Is there proof of the long term stability of the measurement setup for hundreds of cycles as would be needed for aging studies? Does the intensity of the measurement remain high enough over long term cycling as electrolyte is slowly consumed? Without this information, this could be an outlook of such a work, but seems to be misleading as the first paragraph of the introduction.
7. Figure 3b: it would be useful to know which point in the spectral evolution corresponds to the end of charge/beginning of discharge.
8. Figure 4c: the y-axis is unclear. From the text it seems that this is the ratio of the Li-

coordinated peak EC to the Li-coordinated peak of EMC. However, the figure suggests that this is either the uncoordinated peaks or the total ratio. This should be clarified.

9. Figure 5 – the caption says the data was normalized. To what peak? Based on what criteria?

10. The authors suggest that VC forms within the cells and is used as a key justification of the utility of the method. This is surprising, as they note. This result should be verified by another ex-situ measurement technique (not Raman) to verify the increase of the VC concentration (or better its presence in the cells with the VC free electrolytes). Is there any other species that have a Raman signal at similar peak positions that could explain this peak as well?

11. The authors state: "The cell retains ca. 90% of its discharge capacity after 13 cycles (compared to the first-cycle discharge), demonstrating that the insertion of the glass micro-capillary does not affect the electrochemical performance of the cell substantially." A capacity retention of 90% after only 13 cycles does not support the statement that the performance is not substantially affected. This should be revised. As an example (DOI: 10.1149/2.0571910jes), cells cycled to 4.6V with a similar chemistry maintained 90% capacity after 100 cycles, while those cycled to 4.2V showed no fading.

12. Lines 202-203: The authors should explain for the reader why the "EC breathing mode" intensity increases when the LiPF₆ concentration increases. The reader might expect that the intensity should stay the same, but that the peak frequency shifts or a shoulder appears due to coordination. This will be for the general audience unclear.

13. Line 262, The authors refer to the EC breathing mode and state that the intensity does not change. How was this determined: using total integrated intensity, the peak maximum, or ...?

14. Lines 305-307: The authors observe a 5-fold increase in the EC intensity that cannot be explained by any of their other data. Can overall intensity changes in the Raman explain this difference? As noted in the previous point, how does the normalized peak area change?

15. In lines 311 to 313, the authors state that the changes in the EC intensity could possibly be attributed to changes of 0.1% in the refractive index. However, in lines 362-365 in the experimental section, they say the HC-Fibre is robust towards a wide range of refractive indices. How the authors reconcile these different statements?

16. Page 18 line 335 – 338. How were the electrodes printed? What kind of printer? What is the role of the oxalic acid?

17. The electrolyte amount should also be given in the experiment section.

18. Page 10, line 179-180 refers one to Figure S4 for details of the cell construction. This figure is misleading, as it appears to refer to a rectangular pouch. However, it seems from the experimental section that disks were used. Please adjusted the figure to better reflect the cell set. Furthermore, dimensions would greatly help this figure to get a feel for the size of the cell and the size of the probe.

19. More information should be given about the position of the probe in the cell. From figure S4, it seems that the probe is located in the area of the current collectors where there is no electrode coating. Is this the case? Will this have any influence on the measurements?

20. How many cells were measured?

21. Please give the electrode capacity also in mAh cm⁻². How is the full cell balanced?

22. A description of how the Raman spectra should be included in the experimental section.

23. The English should be improved. There are many long sentences that are hard to follow and constructions that are not common in English.

Reviewer #3 (Remarks to the Author):

The manuscript by Miele et al. presents the use of novel hollow-core optical fibre sensor for

operando monitoring of the complex reactions undergoing in Li-ion battery electrolyte. The method reported by authors is very important and will be useful for the researchers in the field. It is particularly valuable that authors employ the full Li-ion cell and not a half cell configuration, which would allow monitoring the crosstalk effect. Authors demonstrate that the probe does not perturb the cell geometry and electrochemistry, and the probe could be applied to monitor commercial Li-ion cells. The reported experimental methodology is sound and will benefit the researchers.

However, a major concern is the interpretation of reported experimental results, specifically Raman spectra. In the experimental details the stated resolution is 40 cm⁻¹, that is unfortunately extremely low in order to support any conclusions related to the Li⁺ solvation dynamics. The studied electrolyte LP57+VC contains four components (EC, VC, LiPF₆ and VC), that have complex spectra and present multiple peaks in several wavenumber intervals, particularly in the 1700 to 1900 cm⁻¹ region. The difference between some of peak positions of the components is lower than the reported resolution of 40 cm⁻¹. Therefore, authors detect (and also assign) less peaks than what is present in the system, which means that proper deconvolution cannot be performed. I will focus on three examples:

1. Region 700-750 cm⁻¹. In figure 2 authors see one very broad peak in this interval, and in the line 148 authors note that there are two peaks PF₆⁻ at 740 and EC skeletal mode at 720 cm⁻¹. However, there should be three distinct peaks in this region, two described by the authors and one more peak at about 725-730 cm⁻¹, corresponding to EC mode when coordinated with Li⁺ (solvating lithium ion) [1-3]
2. Region 800-1000 cm⁻¹. Authors highlight EC breathing at 893 cm⁻¹, and in the supporting information figure 5 they present band variation as a function of lithium concentration. In the lines 207-208 they claim that EC breathing mode intensity doubles when changing concentration from 0.5 to 1M. However, the region marked by authors as EC breathing mode in figure 5 (main text) contains two peaks: EC breathing at 893 cm⁻¹ and EC breathing when coordinated to Li⁺ at 905 cm⁻¹. [1-3] Even VC has a breathing band at the same position but is unlikely to contribute to spectrum because of its very low concentration. Pure EC would only exhibit breathing mode at 893, however when lithium salt is added, another band at 905 is observed. The ratio between those two bands intensities is proportional to the Li⁺ concentration, and that would be a correct way to perform the calibration of Raman response, not just EC breathing mode intensity. The low resolution of 40 cm⁻¹ does not allow to deconvolute two bands properly, and the reported intensity includes mean value of the both peaks.
3. Region 1700-1900 cm⁻¹. This region is a typical for C=O vibrations and is used by authors to conclude on Li-ion solvation dynamics. The reported spectrum in this interval has two broad Raman peaks, which authors deconvolute with 4 peaks: EMC(1750), Li+EMC(1735), Li+EC(1787) and EC/VC(1805). However, EC (not coordinated to Li⁺) has two C=O vibrations: at 1770 cm⁻¹ and 1800 cm⁻¹, [4] and the first one was not included in the performed deconvolution. Once again, because not all the peaks are included, and resolution is low, the extracted data is not reliable.

On the other hand, in the figure 5c, Li⁺ solvation dynamic is tracked using the ratio of EC C=O (~1790-1810 cm⁻¹) and EMC C=O (~1730-1750 cm⁻¹) bands (lines 193-194, 212-214). These two bands, as authors themselves described, correspond to C=O bands of solvents not coordinated to lithium ion, and therefore their ratio does not track the Li-ion solvation, but just the ration between solvent molecules(bulk) in the electrolyte.

Thus, the authors conclusions on the Li-ion solvation dynamics are not supported by experimental evidence. The very low resolution of 40 cm⁻¹, as well as not complete or erroneous Raman bands assignment, does not allow to perform proper band deconvolution.

Therefore, extracted peak intensities cannot be interpreted as reliable data. Furthermore, the authors track the band ratio of bulk solvent molecules instead of those solvating Li-ions.

Also, because of the very low concentration of VC additive, its C=C band signal-to-noise ratio is very low, and its intensity variation lies below the experimental error.

The advancement of the experimental methodology is valuable and can be of importance for future battery research, however the interpretation of results and conclusions require major revision where the authors should either report the data with a higher resolution (at least 2 cm⁻¹) and a proper peak deconvolution; or use a different system to validate their methodology.

References

1. Allen, J. L., Borodin, O., Seo, D. M. & Henderson, W. A. Combined quantum chemical/Raman spectroscopic analyses of Li⁺ cation solvation: Cyclic carbonate solvents—Ethylene carbonate and propylene carbonate. *J. Power Sources* 267, 821–830 (2014).
2. Yang, G. et al. Electrolyte Solvation Structure at Solid–Liquid Interface Probed by Nanogap Surface-Enhanced Raman Spectroscopy. *ACS Nano* 12, 10159–10170 (2018).
3. Mozhzhukhina, N. et al. Direct Operando Observation of Double Layer Charging and Early Solid Electrolyte Interphase Formation in Li-Ion Battery Electrolytes. *J. Phys. Chem. Lett.* 11, 4119–4123 (2020).
4. Fortunato, B., Mirone, P. & Fini, G. Infrared and Raman spectra and vibrational assignment of ethylene carbonate. *Spectrochim. Acta Part A Mol. Spectrosc.* 27, 1917–1927 (1971).

We are highly encouraged by the positive comments from all three referees emphasising: “*new in situ fiber Raman spectroscopy of batteries*” and “*can be published with minor revision*” (Reviewer 1), “*interesting and to my knowledge novel*” and “*can most likely bring interesting results, knowledge and thinking to the field of electrolytes and batteries*” (Reviewer 2), “*very important and will be useful for the researchers in the field*” and the “*experimental methodology is sound and will benefit the researchers*” (Reviewer 3).

Response to Reviewers:

Reviewer #1:

In this paper the authors report a new in situ fiber Raman spectroscopy of batteries. New detailed electrolyte chemistries could be observed. This paper can be published with minor revision.

1) The authors claim that the spectra obtained by the method are background free. An example of spectrum that clearly shows the background level should be presented (where is zero in Y scales?).

> We thank the Reviewer for the positive comments. Indeed, the Raman spectra obtained with our hollow-core fibre spectroscopy method are background free, in the sense that they do not suffer from a large silica background present in conventional fibre-based Raman probes. The baseline level present in our raw spectra is part of the actual Raman response of the infiltrated electrolyte sample (**Fig. R1**). We subtract the baseline level as part of the standard data processing procedures used to prepare the spectra in the manuscript. To clarify this, we have now used the phrases “*silica background-free*” or “*Raman background to due to silica*” Furthermore, we have now included a new figure (Supplementary Figure 6) to explain and illustrate the background correction procedure, shown below (**Fig. R1**):

Fig. R1 | Raman baseline subtraction. Comparison between LP57 +VC 2% wt. Electrolyte before (continuous line) and after (dashed line) baseline subtraction. For each spectrum, the baseline is fitted with an order 3 polynomial curve. A Savitzky–Golay filter (window size 21 points, order 3) is finally applied to smooth the spectra. Counts are referenced to the input power of the pump laser.

Reviewer #2:

The operando method described in the paper is interesting and to my knowledge novel. The method itself, as noted by the authors can most likely bring interesting results, knowledge and thinking to the field of electrolytes and batteries. However, I find that the analysis of the spectra needs to be reevaluated to better validate the claims of the paper. Specific comments are given in the points below.

(1) All of the measurements are displayed in units of counts. How was it ensured that the overall intensity of the spectrum did not change over time? Were the spectra normalised? Is the same trend in the various modes seen if the normalised and integrated peak area is used as the evaluation criteria?

> We thank the Reviewer for this important observation and suggestion. We chose to show the Raman counts per second per unit of the launched optical pump, to highlight the changes in the overall intensity of the spectrum. Such changes reflect real effects in the electrolyte mixture such as bubble formation (resulting in a loss of optical signal) and changes in Li-ion concentration that induce variations in refractive index and, therefore, Raman collection efficiency.

To address the Reviewer's question, we have created baseline-referenced versions of Figure 4 (**Fig. R2**) and Figure 5 (**Fig. R3**). The procedures used to process the raw data were as follows:

1. A baseline was fitted using a third-order polynomial for each raw Raman spectrum and subtracted from the original spectrum (see **Fig R1**). The resulting baseline-subtracted spectra are used throughout the submitted manuscript and shown below in **Fig R2a-c (LHS)**.
2. The baseline-subtracted spectra are subsequently divided by the baseline spectrum in a second series of spectra to test the effect of baseline referencing, as suggested by the Reviewer.

The resulting graphs are shown below for comparison. As suggested by the Reviewer, the integrated area below each Raman band was used as an evaluation criterium.

Fig. R2 | Original Figure 4 (left) baseline-subtracted data, and (right) the same data with baseline-subtraction and -referencing (as discussed above). The dashed line on figure b indicates the time point where the spectra deconvolution in a was performed.

Fig. R3 | Original Figure 5 (left) baseline-subtracted data, (right) same data, baseline-subtracted and -referenced.

From these data, it can be seen that the baseline referencing procedure does not introduce noticeable changes in the EC/EMC C=O stretch ratio (**Fig. R2**). However, referencing (dividing by) the absolute counts to the baseline spectrum does introduce some differences in the time traces in **Fig. R3**: First, the baseline-referenced vinylene peak stays constant throughout the cycle, rather than showing a slight increase. This small difference is likely due to the additional noise introduced via the referencing operation. What is clear, though, is that the vinylene C=C stretch signal does not disappear during the first cycle: we note that this remains an unexpected result, as literature studies have suggested the consumption of VC during cycling (*Electrochimica Acta* **47**, 1423–1439 (2002)). The assignment of the vinylene peak is discussed below in the context of IR data collected for a second cell. Second, the baseline-referenced EC breathing mode shows an additional spike, around 7000 sec into the cycle, but the overall behaviour of this peak is otherwise essentially identical to the unreferenced data. The observation of the spike underlines the comment in response to Reviewer 1: namely, that the baseline level present in our raw spectra is part of the actual Raman response of the infiltrated electrolyte sample. Other responses such as fluorescence also contribute to this background signal, and the rapid jump and decrease in overall intensity remain a subject of investigation (see below).

(2) The Raman resolution of 10 and 40 cm^{-1} is much lower than normally used in electrolyte studies (1-4 cm^{-1}). What implications does this have on the data analysis? How does this impact the separation of Li-coordinated and non-Li-coordinated peaks of the various species? > This is an important point. The primary aim of our proof-of-principle study was to simultaneously monitor a broad spectral region containing multiple Raman bands of interest. This was achieved by using a diffraction grating with a relatively low density (300 grooves / mm), resulting in a broad detection range between 644-2344 cm^{-1} (see methods section). We chose a slit width of 190 μm to enable imaging of the light distribution across the hollow fibre core. This was done by imaging the zeroth-order diffraction and can be achieved without making mechanical adjustments to the slit width (**Fig. 1c**). The resulting spectral resolution for this configuration was 27 cm^{-1} for the band centred around 900 cm^{-1} . We agree with the Reviewer that the obtained spectral resolution is too low to fully resolve individual peaks within some of the bands (for example, the EC/EMC band around 900 cm^{-1}). The spectral resolution is easily obtained using the online tool provided by Princeton instruments: <https://www.princetoninstruments.com/learn/calculators/grating-dispersion> (see also **Fig. R4**)

Detector	ProEM:1024B	Center Wavelength (nm)	845	Enter		
Hor. pixels	1024	Vert. pixels	1024	Order		1
Pixel Size μm	13	CCD width/mm	13.312	Select Excitation Laser		785 nm
Linewidth, pixels	4.615	Laser wavelength (nm)	785			
Spectrograph	SP-2300					
Focal Length(mm)	300	Aperture ratio	f/3.9			
Slit width	200 μm	Dispersion nm/mm	10.25			Grating rotation ($^\circ$)
		Dispersion nm/pixel	0.133			7.55
Grating	300 l	Center(cm-1)	905			Slit width μm :
		Range (nm)	136			200
Grooves/mm	300	Spacing/nm	3333	Short Wavelength (nm)		776
Grating size/mm	68 x 68, 68 x 84 opt.	Next resolved wavelength	847	Long Wavelength (nm)		913.22
		Low Raman shift (cm-1)	-135	Next resolved wavenumber		11805
nm to eV		CCD Resolution (nm)	2.0501	High Raman shift (cm-1)		1789
		Resolution (cm-1)	28.64			

Detector	ProEM:1024B	Center Wavelength (nm)	845	Enter		
Hor. pixels	1024	Vert. pixels	1024	Order		1
Pixel Size μm	13	CCD width/mm	13.312	Select Excitation Laser		785 nm
Linewidth, pixels	4.923	Laser wavelength (nm)	785			
Spectrograph	SP-2300					
Focal Length(mm)	300	Aperture ratio	f/3.9			
Slit width	100 μm	Dispersion nm/mm	1.9			Grating rotation ($^\circ$)
		Dispersion nm/pixel	0.025			31.69
Grating	1200 l	Center(cm-1)	905			Slit width μm :
		Range (nm)	25			100
Grooves/mm	1200	Spacing/nm	833	Short Wavelength (nm)		832
Grating size/mm	68 x 68, 68 x 84 opt.	Next resolved wavelength	845	Long Wavelength (nm)		857.65
		Low Raman shift (cm-1)	725	Next resolved wavenumber		11831
nm to eV		CCD Resolution (nm)	0.19	High Raman shift (cm-1)		1079
		Resolution (cm-1)	2.66			

Detector	ProEM:1024B	Center Wavelength (nm)	845	Enter		
Hor. pixels	1024	Vert. pixels	1024	Order		1
Pixel Size μm	13	CCD width/mm	13.312	Select Excitation Laser		785 nm
Linewidth, pixels	4.923	Laser wavelength (nm)	785			
Spectrograph	SP-2300					
Focal Length(mm)	300	Aperture ratio	f/3.9			
Slit width	10 μm	Dispersion nm/mm	1.9			Grating rotation ($^\circ$)
		Dispersion nm/pixel	0.025			31.69
Grating	1200 l	Center(cm-1)	905			Slit width μm :
		Range (nm)	25			10
Grooves/mm	1200	Spacing/nm	833	Short Wavelength (nm)		832
Grating size/mm	68 x 68, 68 x 84 opt.	Next resolved wavelength	845	Long Wavelength (nm)		857.65
		Low Raman shift (cm-1)	725	Next resolved wavenumber		11832
nm to eV		CCD Resolution (nm)	0.1216	High Raman shift (cm-1)		1079
		Resolution (cm-1)	1.7			

Fig R4 | Spectrometer resolution for different spectrometer configurations. (a) broadband, low resolution setting: 300 g/mm grating, 200 μm slit, (b) narrowband, high-resolution setting: a 1200 g/mm grating, 100 μm slit, (c) highest-resolution setting: 1200 g/mm grating, 10 μm slit. Screen shots taken from: <https://www.princetoninstruments.com/learn/calculators/grating-dispersion>

To demonstrate that our fibre probe does not limit the resolution, we have now performed additional high-resolution Raman measurements using a reduced slit width of 90 μm and 1200 grooves/mm grating. This configuration results in a resolution $\sim 2.4 \text{ cm}^{-1}$ over a reduced spectral range of 650-1000 cm^{-1} (see **Fig. R4**). The resulting spectra on an LP57 +2% VC electrolyte sample in **Fig. R5** (also included as a new Supplementary Figure 7) demonstrate a clear separation between the Li-coordinated and non-Li-coordinated peaks. For a detailed discussion of these data, see our Response to Reviewer 3.

Fig. R5 | High-resolution Raman spectra. Comparison between high- and low-resolution ex-situ Raman spectra on non-cycled electrolyte, obtained using low (300 g/mm, 190 μm slit width, 27 cm^{-1} resolution, black, **a**) and high (1200 g/mm, 90 μm slit width, 2.4 cm^{-1} resolution, orange/green, **b** and **c**) resolution settings. Raman vibrations of interest for Li-ion solvation studies are highlighted in the high-resolution Raman scans (**b** and **c**). The high-resolution spectra in **b** and **c** have also been added as orange dashed and green lines in **a**. This Figure is now included as new Supplementary Figure 7.

(3) Do the authors expect any effects due to convection as a result of removing and reintroducing the electrolyte in the cell? Do the authors expect any effect due to the reduction of the electrolyte amount by 25% during measurement? What is the electrolyte excess?

> The total electrolyte volume inserted into the cell is 100 μL , the effective volume trapped between the two electrodes (electrode porosity 0.3) is ~ 25 μL , resulting in a 75 μL excess volume. During our measurements, 30% (25 μL) of electrolyte is extracted. After the Raman analysis, the entire sample is reinserted. We have analysed whether the temporary reduction in electrolyte volume during measurements affects the electrochemical behaviour of the cell. **Figure R6** shows a typical dQ/dV trace under repeated sampling. Only minimal changes in the dQ/dV are observed during the sampling cycle, indicating that the method does not affect the electrochemical performance of the cell. (

4) Page 2, Line 29-30 – is there a reason to believe that degradation of high-energy cathodes is higher in electric vehicle batteries than in batteries for other applications with the same chemistry?

> We agree with the Reviewer that this statement is not limited to a particular user application and have therefore removed “electric vehicle” from the text. However, we note that the Ni-rich high-energy cathodes are predominantly used in EV batteries and not portable electronics.

Fig. R6 | Electrochemistry during sampling. dQ/dV plot showing minimal modulation (arrows) due to the reduction of the electrolyte volume during each sampling/infusion cycle.

(5) Page 4, line 83-84: what challenges exactly are you referring to in this passage? Furthermore does this refer to all spectroscopy methods or only Raman spectroscopy methods.

> In lines 83-84 we meant to refer to the challenges and limitations in existing operando methods based often on half-cells with geometries discussed in lines 53-62. These challenges include:

1. The need for an access window in the cell, resulting in geometries that do not capture the complex cross-talk between cathode, electrolyte and anode processes. Often the window is placed on the back of the electrode (i.e., on the opposite side to that facing the separator), and in many cases, the cells are cycled against Li metal (i.e., they are so-called half-cells) and cross-talk mechanisms will be different.
2. The electrolyte volume-to surface area in many in-situ cells for optical analysis of batteries exceeds that used in commercially used cells.
3. The difficulty to reproduce real operating conditions (battery pack).

To make this reference more explicit, we have rephrased the wording of lines 83-84 to: *“to enable operando spectroscopy on full-cell batteries, thus overcoming the challenges and limitations facing many cells designed for optical measurements.”*

(6) Lines 28-43: The entire first paragraph of the manuscript is dedicated to the discussion of degradation and aging mechanisms. However, in the last paragraph of the introduction, the authors state that the proof of concept cell is used to monitor electrolyte chemistry during the first cycle during SEI formation. Is there proof of the long term stability of the measurement setup for hundreds of cycles as would be needed for aging studies? Does the intensity of the measurement remain high enough over long term cycling as electrolyte is slowly consumed? Without this information, this could be an outlook of such a work, but seems to be misleading as the first paragraph of the introduction.

> Some of the degradation effects mentioned in the introduction only become prominent after >100 cycles, so it is still instructive to highlight these effects, as our manuscript demonstrates the ability to track the Raman bands relevant to these processes.

We agree with the Reviewer that the ability to measure Raman spectra after multiple cycles is essential for future tests over tens to hundreds of cycles. We have therefore included further cycling data (over 14 cycles) in **Figure 6**. In these experiments, the detection fibre was temporarily disconnected from the cell, allowing it to be cycled multiple times before being reconnected to the Raman setup. This procedure did not cause any differences in optical signal or Raman counts.

Regarding potential electrolyte consumption: as the sampling system is completely sealed, in the sense that any electrolyte extracted from the cell is reinserted after the measurement, the measurement does not introduce any additional electrolyte consumption beyond what would be expected in regular cycling experiments. These additional data clearly illustrate that the method will be capable of periodically monitoring cells in ageing studies over 100s of cycles. In such experiments, a single Raman measurement per cycle would generally suffice.

(7) Figure 3b: it would be useful to know which point in the spectral evolution corresponds to the end of charge/beginning of discharge.

> We agree that this would be helpful, and we have now highlighted the spectra corresponding to the start and the end of the charge cycle in our updated Fig. 3b (see Fig. R7 below).

Fig. R7 | updated Figure 3b – also included in manuscript.

(8) Figure 4c: the y-axis is unclear. From the text it seems that this is the ratio of the Li-coordinated peak EC to the Li-coordinated peak of EMC. However, the figure suggests that this is either the uncoordinated peaks or the total ratio. This should be clarified.

> We agree that the caption in Figure 4c should be improved. The data show the ratio between the counts in the EC C=O band (solvated and un-solvated) and the EMC C=O band (solvated and un-solvated). To clarify this in the manuscript, we have rephrased the wording in the caption of Fig. 4c to: “trend of C=O band intensity ratio between EC C=O (solvated and un-solvated 1782-1817 cm^{-1}) and EMC C=O (solvated and un-solvated bands, 1730-1765 cm^{-1}).” We have also clarified the corresponding passage in the manuscript.

(9) Figure 5 – the caption says the data was normalised. To what peak? Based on what criteria?

> The Reviewer’s comment helped us catch a typo in the caption of Fig. 5: the data shown here have been baseline-subtracted but have not been normalised. Therefore, we have changed the description in the caption from “normalised” to “baseline-subtracted”.

(10) The authors suggest that VC forms within the cells and is used as a key justification of the utility of the method. This is surprising, as they note. This result should be verified by another ex-situ measurement technique (not Raman) to verify the increase of the VC concentration (or better its presence in the cells with the VC free electrolytes). Is there any other species that have a Raman signal at similar peak positions that could explain this peak as well?

> We agree with the Reviewer that the surprising occurrence of the vinylene Raman band during cycling (Figure 6) is a good example of how we can now study chemical changes in full cells. In our earlier manuscript, we attributed this band to vinylene carbonate. However, we agree with the Reviewer that the nature of the species needed to be confirmed using a different ex-situ technique. We have therefore carried out additional experiments using electrolyte extracted from a 10-times cycled NMC811 coin cell (electrolyte extraction procedure as in *J. Electrochem. Soc.* **165**, A2732 (2018)), that was prepared with LP57 electrolyte without VC (as suggested by the Reviewer). We analysed the extracted electrolyte with our Raman technique as well as with ex-situ ATR FT-IR spectroscopy. We compared the resulting spectra with those of fresh LP57 and LP57 + 2% VC electrolytes. The Raman spectra for the cycled cell (Fig. R8a-b) clearly showed the appearance of a Raman band at $\sim 1628\text{ cm}^{-1}$. This vinylene band is also present in the operando data in our Manuscript (Fig. 6c), and we previously identified it as belonging to VC.

Fig. R8 | Ex-situ Raman and ATR FT-IR spectra. A coin cell was assembled from the same electrode materials as in the main text experiments (NMC 811 cathode, Graphite). A 1x 17 mm dia. glass fibre separator (GF/A, Whatman) was placed between 2x 16 mm dia. Celgard separators, which are soaked with 140 μL of LP57 electrolyte with no VC added. The cell was cycled 10 times between 2.5-4.2 V at a C/20 rate assuming a practical capacity for NMC811 of 185 mAh/g. At cycle 10, the cell was opened under Ar atmosphere, and electrolyte was extracted by centrifuging in falcon tubes (as in *J. Electrochem. Soc.* **165**, A2732 (2018)). Raman spectra (a,b) were measured with the HC-fibre probe. ATR FT-IR spectra (c) were collected using a Shimadzu IRTracer-100 FT-IR spectrometer with QATR-10 ATR crystal. This Figure has also been included as new Supplementary Figure 10.

However, the ATR FT-IR data for the cycled sample (**Fig. R8c**) differ from those of pristine LP57 + 2% VC. For example, the absorption peak at 1830 cm^{-1} in the LP57+ 2% VC spectrum, belonging to the VC C=O stretch bond, is absent in the cycled LP57 IR spectrum. Instead, the cycled LP57 sample features a strong IR absorption band centred around 1620 cm^{-1} in both the IR and Raman spectra, consistent with a C=C bond in an asymmetric molecule. This absorption band thus suggests the presence of linear (rather than cyclic) reduced vinylene species such as lithium vinylene carbonate and lithium vinylene alkoxide. Such linear species can form as a soluble product of SEI or VC degradation at the anode (*J. Electrochem. Soc.* **151**, A1659 (2004)). More recently, reduced VC species have been proposed (lithium vinylene di-carbonate (LVD) and lithium divinylene di-carbonate (LDVD)), which give rise to IR bands close to 1600 cm^{-1} (*Journal of Power Sources* **427**, 77–84 (2019)).

Some of these species are likely formed in the first cycle. Therefore, we have included the ATR FT-IR data as a new Supplementary Figure 10 in the revised manuscript to clarify this critical point. In addition, we refer to “vinylene species” throughout the text (rather than to VC), and we have adjusted the discussion to our new insights.

(11) The authors state: “The cell retains ca. 90% of its discharge capacity after 13 cycles (compared to the first-cycle discharge), demonstrating that the insertion of the glass micro-capillary does not affect the electrochemical performance of the cell substantially.” A capacity retention of 90% after only 13 cycles does not support the statement that the performance is not substantially affected. This should be revised. As an example (DOI: 10.1149/2.0571910jes), cells cycled to 4.6V with a similar chemistry maintained 90% capacity after 100 cycles, while those cycled to 4.2V showed no fading.

> We thank the Reviewer for this comment and the useful comparison to the excellent capacity retention over 100 cycles reported in cells similar to ours by Laszczynski *et al.* (*J. Electrochem. Soc.* **166** A1853 (2019)). We have carefully studied the paper by Laszczynski *et al.*, and have concluded that, while demonstrating very low capacity loss in later cycles, their analysis only considers capacity loss occurring **after the first 5 formation cycles**, as seen in **Fig R9**. During the first 5 cycles, Laszczynski *et al.* report a 10% capacity loss (from 185 mAh g to 165 mAh g for a C/2 C-rate cycled to 4.2V), which is identical to the 10% loss we observed during the first 13 cycles to 4.2V in our experiment.

We find it instructive to compare the capacity loss of our cells during formation cycles to literature values and have therefore included a new reference to the paper by Laszczynski *et al.*. However, we appreciate that our data over the first 13 cycles does not predict the long-term cycling behaviour of our cell. This is something we will investigate in future long-term cycling experiments.

Fig. R9 | Electrochemical performance of Graphite/NCM811 cells cycled at different cut off voltages of 4.2 and 4.6 V, respectively. a) Galvanostatic cycling at a C-rate of C/2 **after formation cycles** at C/20 (1), C/10 (2-3), and C/5 (4-5) (reproduced from: *J. Electrochem. Soc.* **166** A1853 (2019)).

(12) Lines 202-203: The authors should explain for the reader why the “EC breathing mode” intensity increases when the LiPF₆ concentration increases. The reader might expect that the intensity should stay the same, but that the peak frequency shifts or a shoulder appears due to coordination. This will be for the general audience unclear.

> We thank the Reviewer for this excellent comment. Upon close inspection, the EC breathing mode band **Figure 5a** indeed shows a slight shift to lower frequency during charging. The Reviewer is correct that this is due to coordination of the underlying peaks, which cannot be resolved in the broadband, low-resolution Raman spectra showed here. However, the underlying peak is visible in the new high-resolution spectra (see **Fig R5/** response to comment 2). The potential reasons for the increase in EC breathing mode intensity, including the effect of the solvation structure, are discussed in detail the new manuscript:

*“The EC breathing mode shows a significant evolution in intensity during the formation cycle (**Fig. 5d**), consistent with the trend in the ratio of EC/EMC C=O stretch (**Fig 4c**). During the first cycle, we observe an initial five-fold increase in the EC breathing mode. However, we note that the operando Raman data in cycle 7 (**Fig. 6b**) do not show significant EC-band fluctuations, supporting the hypothesis that these changes are related to the formation cycle. To check how the first-cycle increase in EC-signal is affected by variations in Li⁺ concentration, calibration experiments were performed in which the Li⁺ concentration was varied from 0.5 to 1 M, resulting in an at most two-fold increase in Raman intensity (Supplementary Figure 8). The change in intensity of the EC breathing mode (880-915 cm⁻¹) may be linked to a modification in solvation structure of Li⁺-O=C of EC.^{24,63} It is also consistent with earlier FT-IR studies in a half-cell configuration,⁹ that showed that the peak position and intensity of the EC C=O IR absorption band is affected by the Li⁺ concentration, and that EC undergoes a progressive dehydrogenation correlated with the cathodic potential.⁹ Furthermore, changes in Li⁺ solvation will increase the refractive index of the electrolyte by typically 0.001 per Li⁺ molar concentration, as was demonstrated with different salts and salt concentrations.⁶⁹ This would shift the transmission window of the fibre by up to ~4 nm (Supplementary Figure 3b), and slightly affect the amount of Raman pump light propagating through the hollow-core fibre.⁴⁰ However, even when taking into account all the above effects, the five-fold increase in EC-Raman signal cannot be fully explained by changes in Li⁺ concentration alone, suggesting that other unknown effects may play a role.”*

(13) Line 262, The authors refer to the EC breathing mode and state that the intensity does not change. How was this determined: using total integrated intensity, the peak maximum, or ...?

> The EC breathing mode intensity plotted in **Fig. 6b** was obtained by integrating the counts within the frequency band between 848-943 cm⁻¹. This band comprises the Li-solvated EC breathing mode (906 cm⁻¹), the un-solvated EC-breathing mode (896 cm⁻¹), and an EMC vibrational mode at 930 cm⁻¹, all of which are visible in the high-resolution Raman spectra in **Fig. R5b**.

(14) Lines 305-307: The authors observe a 5-fold increase in the EC intensity that cannot be explained by any of their other data. Can overall intensity changes in the Raman explain this difference? As noted in the previous point, how does the normalised peak area change?

> We thank the Reviewer for this question, which is directly related to their comment #1. In our reply to comment #1, we referenced our Raman data to the baseline level to eliminate overall intensity fluctuations (**Fig R1**). Regarding peak area changes: our data analysis does use the integrated area within a band near the peak, as suggested by the Reviewer. Our analysis should therefore be relatively insensitive to small shifts in the peak frequency. The referenced data (**Fig R3**) indeed still shows a significant fluctuation in the EC intensity.

(15) In lines 311 to 313, the authors state that the changes in the EC intensity could possibly be attributed to changes of 0.1% in the refractive index. However, in lines 362-365 in the experimental section, they say the HC-Fibre is robust towards a wide range of refractive indices. How the authors reconcile these different statements?

> We agree that we should explain the guidance properties of the fibre and their robustness in terms of refractive index changes in more detail. Therefore, we have created a new figure (**Fig R10a**) to show how the anti-resonance and resonance wavelengths depend on wall thickness. For the current fibre, with a core wall thickness of around 425 nm, the first anti-resonance guidance band is centred at 725nm. This band is close to the wavelength of the Raman laser, and reasonable guidance is expected for a (typically 30%) broad range around this centre wavelength.

Figure **R10b** shows how the guidance band position depends on the refractive index of the infiltrated liquid. We see that the band blueshifts by around 40 nm for a refractive index increase of $\Delta n = 0.01$. While this shift should not significantly alter the guidance properties of the fibre, some changes in transmitted power may still occur (see Supplementary Fig. 2). For clarity, we have included **Figure R10** in the revised manuscript as a new Supplementary Fig. 3.

Fig. R10| Anti-resonance and resonance spectral position. (a) Position of the anti-resonances (green lines) and resonances (red lines) of the fibre, as predicted by the ARROW model (Opt. Lett. **27**, 1592–1594 (2002).) for $n_{core} = 1.39$ and $n_{glass} = 1.454$. The shaded band indicates a $\sim 30\%$ spectral width of the first guidance band that is typical for ARROW waveguides. The vertical dashed line indicates the average wall thickness of the capillaries in the fibre core (425 nm), the horizontal shaded band indicates the spectral region of interest (785 – 910 nm). The Raman band lies well above the first loss resonance ($\lambda = 380 \text{ nm}$), and therefore reasonable guidance is expected. (b) Dependence of the anti-resonance position on liquid-core refractive index. The anti-resonance position shifts by -4000 nm per refractive index unit (RIU). This Figure has also been included as new Supplementary Fig. 3.

(16) Page 18 line 335 – 338. How were the electrodes printed? What kind of printer? What is the role of the oxalic acid?

> To clarify the electrode printing process as suggested, we have added a detailed description to the Methods section. After “... Cell Analysis, Modelling, and Prototyping (CAMP) facility” we add:

“The electrodes were coated in a climate-controlled dry room with a dew point less than $42 \text{ }^\circ\text{C}$ ($<100 \text{ ppm}$ moisture) at a rate of 0.4 m min^{-1} using a pilot-scale semi-automatic coater (A-PRO Co.).” After “...on 20 μm thick aluminium foil.” we have added: “Electrodes were calendared at 0.5 m min^{-1} using a heated ($80 \text{ }^\circ\text{C}$) two-roller hydraulic-driven roll press (A-PRO Co.) to 30 % porosity.”.

The oxalic acid in the slurry partially etches the Cu current collector by removing surface oxides and thus improves graphite adhesion to the substrate (*Journal of Power Sources* **195**, 7090 (2010)).

(17) The electrolyte amount should also be given in the experiment section.

> The amounts of electrolyte were 100 μL for the operando measurements with circular electrodes in Figs. 3-5, and 260 μL for the multi-cycle experiments with square electrodes in **Fig. 6**. We have included these values in the Methods section.

(18) Page 10, line 179-180 refers one to Figure S4 for details of the cell construction. This figure is misleading, as it appears to refer to a rectangular pouch. However, it seems from the experimental section that disks were used. Please adjusted the figure to better reflect the cell set. Furthermore, dimensions would greatly help this figure to get a feel for the size of the cell and the size of the probe.

> We thank the Reviewer for this suggestion and agree that an experimental image with dimensions will help the reader. Therefore, we have revised Supplementary Figure 3 (Supplementary Figure 5 in new manuscript) to include a 3D schematic and a photograph showing the cell assembly method. The main experiments in **Figs. 3-5** used circular electrodes with a diameter of 14 mm (area 1.54 cm^2). The multi-cycle experiments in **Figure 6** used square electrodes (area 4 cm^2).

Figure R11 | Li-ion cell stack geometry and assembly. (a-b) The hollow-core fibre is embedded in the electrolyte compartment, protected by two 25 μm -thick layers of monolayer PE polymer separator (MTI). (c) Cross-section of the hollow-core fibre position within the cell. The outer diameter of the fibre is 174 μm . (d) Photograph of cell assembly using square electrodes with a surface area of 4 cm^2 (as used for the multi-cycle experiment in **Fig. 6**). An identical assembly method (but with circular electrodes) was used for the data in **Figs. 3-5**.

(19) More information should be given about the position of the probe in the cell. From figure S4, it seems that the probe is located in the area of the current collectors where there is no electrode coating. Is this the case? Will this have any influence on the measurements?

> As the Reviewer suggests, we have updated Supplementary Figure 5 to better show the fibre probe position within the cell (see Figure **R11** above). Laterally, the fibre was placed between the two electrode tabs i.e., in the region where there is a coating on both electrodes. The probe was inserted until it reached the edge of the electrode-covered area.

(20) How many cells were measured?

> Operando Raman measurements were performed on five different cells. In the new manuscript, we have included ex-situ data obtained from two additional experiments in Supplementary Figure 9.

(21) Please give the electrode capacity also in mAh cm⁻². How is the full cell balanced?

> To address this comment, we have included the following wording in the Methods section of the manuscript: *“The cathode and anode electrodes had active mass loadings of 8.21 mg_{NMC} cm⁻² (corresponding to ~1.52 mAh cm⁻² based on 185 mAh g⁻¹_{NMC}) and 5.83 mg_{Gr} cm⁻² (~1.92 mAh cm⁻² based on 330 mAh g⁻¹_{Gr}), respectively. The n:p ratio (i.e., the fraction of capacity on the anode (n) relative to the cathode (p)) for these electrodes is in the range 1.16-1.26.”*

(22) A description of how the Raman spectra should be included in the experimental section.

> To clarify the data processing procedure, we have included a new Supplementary Figure 6, also shown as **Fig. R1** (response to Reviewer 1). For each spectrum, a baseline was obtained by fitting an order 3 polynomial fit which was then subtracted from the data. Finally, a Savitzky–Golay filter (window size 21 points, order 3) was applied to smooth the spectra.

(23) The English should be improved. There are many long sentences that are hard to follow and constructions that are not common in English.

> We thank the Reviewer for the suggestion on how to improve the clarity of our manuscript. We have carefully edited the text to obtain shorter and clearer sentence constructions where possible.

Reviewer #3:

The Manuscript by Miele et al. presents the use of novel hollow-core optical fibre sensor for operando monitoring of the complex reactions undergoing in Li-ion battery electrolyte. The method reported by authors is very important and will be useful for the researchers in the field. It is particularly valuable that authors employ the full Li-ion cell and not a half cell configuration, which would allow monitoring the cross-talk effect. Authors demonstrate that the probe does not perturb the cell geometry and electrochemistry, and the probe could be applied to monitor commercial Li-ion cells. The reported experimental methodology is sound and will benefit the researchers.

(1) However, a major concern is the interpretation of reported experimental results, specifically Raman spectra. In the experimental details the stated resolution is 40 cm⁻¹, that is unfortunately extremely low in order to support any conclusions related to the Li⁺ solvation dynamics. The studied electrolyte LP57+VC contains four components (EC, VC, LiPF₆ and VC), that have complex spectra and present multiple peaks in several wavenumber intervals, particularly in the 1700 to 1900 cm⁻¹ region. The difference between some of peak positions of the components is lower than the reported resolution of 40 cm⁻¹. Therefore, authors detect (and also assign) less peaks than what is present in the system, which means that proper deconvolution cannot be performed. I will focus on three examples:

> We thank the Reviewer for the highly positive comments on the validity and relevance of our operando detection method. In addition, the Reviewer makes an excellent comment about the limited spectral resolution used in the presented operando measurements. A similar comment was made by Reviewer 2 (comment 2). We (partially) repeat the response to Reviewer 2 here for convenience:

The primary aim of our proof-of-principle study has been to simultaneously monitor a broad spectral region containing multiple Raman bands of interest. This was achieved using a diffraction grating with a relatively low density (300 grooves / mm), resulting in a broad detection range between 644-2344 cm^{-1} (see methods session). A slit width of 190 μm was chosen to enable imaging of the light distribution across the fibre's hollow core using the zeroth-order diffraction peak, without the need for mechanical adjustments to the slit width (**Fig. 1c**). The resulting spectral resolution for the band centred around 900 cm^{-1} was 27 cm^{-1} . We agree with the Reviewer that the obtained spectral resolution is too low to fully resolve individual peaks within some bands, including the EC/EMC band around 900 cm^{-1} . To demonstrate that our fibre probe does not limit the resolution, we have performed additional high-resolution Raman measurements on an LP57 +2% VC electrolyte sample using a reduced slit width of 90 μm and 1200 grooves/mm grating. This configuration results in a resolution of 2.4 cm^{-1} (see the response to Reviewer 2, **Fig. R2**), over a reduced spectral range of 650-1000 cm^{-1} (see **Fig. R12**). Below, we discuss the new high-resolution data in the three Raman bands mentioned by Reviewer 3.

Fig. R12 | high-resolution Raman spectra. Comparison between high- and low resolution ex-situ Raman spectra on non-cycled electrolyte, obtained using low (300 g/mm, 190 μm slit width, 27 cm^{-1} resolution, black, **a**) and high (1200 g/mm, 90 μm slit width, 2.4 cm^{-1} resolution, orange/green, **b** and **c**) resolution settings. Raman vibrations of interest for Li-ion solvation studies are highlighted in the high-resolution Raman scans (**b** and **c**). The high-resolution spectra in **b** and **c** have also been added as orange dashed and green lines in **a**. This Figure has also been included as new Supplementary Figure 7.

(2) **Region 700-750 cm⁻¹**. In figure 2 authors see one very broad peak in this interval, and in the line 148 authors note that there are two peaks PF6⁻ at 740 and EC skeletal mode at 720 cm⁻¹. However, there should be three distinct peaks in this region, two described by the authors and one more peak at about 725-730 cm⁻¹, corresp. to EC mode when coordinated with Li⁺ (solvating lithium ion) [1–3]
> We agree with the Reviewer that a third peak should be present in this region. Indeed, three separate peaks are visible in our new high-resolution spectrum in **Fig. R12b** (also included as new Supplementary Figure 7). This also includes a peak at 725 cm⁻¹, corresponding to the EC skeleton mode when coordinated with Li⁺.

(3) **Region 800-1000 cm⁻¹**. Authors highlight EC breathing at 893 cm⁻¹, and in the supporting information figure 5 they present band variation as a function of lithium concentration. In the lines 207-208 they claim that EC breathing mode intensity doubles when changing concentration from 0.5 to 1M. However, the region marked by authors as EC breathing mode in figure 5 (main text) contains two peaks: EC breathing at 893 cm⁻¹ and EC breathing when coordinated to Li⁺ at 905 cm⁻¹. [1–3] Even VC has a breathing band at the same position but is unlikely to contribute to spectrum because of its very low conc.. Pure EC would only exhibit breathing mode at 893, however when lithium salt is added, another band at 905 is observed. The ratio between those two bands intensities is proportional to the Li⁺ concentration, and that would be a correct way to perform the calibration of Raman response, not just EC breathing mode intensity. The low resolution of 40 cm⁻¹ does not allow to deconvolute two bands properly, and the reported intensity includes mean value of the both peaks.
> The Reviewer correctly mentions that the broad peak around 900 cm⁻¹ comprises two peaks related to the EC breathing mode 893 cm⁻¹ and the EC breathing when coordinated to Li⁺ at 905 cm⁻¹. These two underlying peaks become visible in the high-resolution spectrum in **Fig. R12b**. (also included as new Supplementary Figure 7). These individual peaks are not resolved in our broadband operando spectra. However, we note that the EC breathing mode band Figure 5a does slightly shift during the cycle, which could reflect the change in the ratio between the underlying 906 cm⁻¹ and 893 cm⁻¹ peaks, as suggested by the Reviewer. Thus, we agree with the Reviewer that our method opens up great opportunities for future experiments on Li⁺ solvation dynamics using high-resolution spectra such as the one presented in **Fig. R12b**. However, we feel these fall outside the scope of the current proof-of-principle demonstration of operando Raman spectroscopy on full-cell batteries.

(4) **Region 1700-1900 cm⁻¹**. This region is a typical for C=O vibrations and is used by authors to conclude on Li-ion solvation dynamics. The reported spectrum in this interval has two broad Raman peaks, which authors deconvolute with 4 peaks: EMC(1750), Li+EMC(1735), Li+EC(1787) and EC/VC(1805). However, EC (not coordinated to Li⁺) has two C=O vibrations: at 1770 cm⁻¹ and 1800 cm⁻¹, [4] and the first one was not included in the performed deconvolution. Once again, because not all the peaks are included, and resolution is low, the extracted data is not reliable.

> In the deconvolution of the spectrum shown in Fig. 4a, we have used the 4 strongest Raman bands, including the EC line at 1804 cm⁻¹ [4] and have, for clarity, omitted weaker lines such as the EC C=O peak at 1773 cm⁻¹ [4]. We agree that a full deconvolution of this spectral region requires a high-resolution spectrum. Our new high-resolution spectrum for this range (**Fig. R12c**) indeed provides a clear separation between the different solvated and un-solvated peaks within this band.

(5) On the other hand, in the figure 5c, Li⁺ solvation dynamic is tracked using the ratio of EC C=O (~1790-1810 cm⁻¹) and EMC C=O (~1730-1750 cm⁻¹) bands (lines 193-194, 212-214). These two bands, as authors themselves described, correspond to C=O bands of solvents not coordinated to lithium ion, and therefore their ratio does not track the Li-ion solvation, but just the ration between solvent molecules(bulk) in the electrolyte.

Thus, the authors conclusions on the Li-ion solvation dynamics are not supported by experimental evidence. The very low resolution of 40 cm⁻¹, as well as not complete or erroneous Raman bands assignment, does not allow to perform proper band deconvolution. Therefore, extracted peak intensities cannot be interpreted as reliable data. Furthermore, the authors track the band ratio of bulk solvent molecules instead of those solvating Li-ions.

> We thank the Reviewer for this comment, which helped us realise that the caption we provided for **Figure 4c** was not clear. What is plotted here is the total ratio between the counts in the EC C=O band (solvated and un-solvated) to the counts in the EMC C=O band (solvated and un-solvated). To clarify this in the manuscript, we have rephrased the wording in the caption of **Fig. 4c** to:

“trend of C=O band intensity ratio between EC C=O (solvated and un-solvated, 1782-1817 cm⁻¹) and EMC C=O (solvated and un-solvated bands, 1730-1765 cm⁻¹).”

Our ratiometric Raman data in **Fig. 4c** demonstrates that, during cycling, the intensity within the EC C=O band reacts differently to Li-solvation compared to the EMC C=O band. Possible reasons for the observed differences are:

1. There will be a solvation-induced shift of the centre position of each Raman band, which in turn changes the fraction of the signal that lies within each fixed integration window.

2. The intensity of the EC C=O band may increase more strongly upon Li-solvation, compared to the EMC C=O band. We note that similar Raman intensity increases with Li-solvation during cycling were observed for the EC-breathing mode in the 800-1000 cm⁻¹ region (**Fig. 5c**).

While our broadband operando Raman spectra do not have a sufficiently high resolution to fully deconvolute the peaks in the 1700-1900 cm⁻¹ region, we do observe clear effects of Li-solvation in this band. Furthermore, the broadband nature of our operando measurement uniquely enables us to link the behaviour in the 1700-1900 cm⁻¹ region with that of the simultaneously monitored EC-breathing mode in the 800-1000 cm⁻¹ region. At the same time, our high-resolution data (**Fig. R12**) highlight the potential of our technique to also track specific Raman bands with high-resolution spectroscopy in future experiments.

(6) Also, because of the very low concentration of VC additive, its C=C band signal-to-noise ratio is very low, and its intensity variation lies below the experimental error.

> We appreciate that the vinylene Raman band is not the strongest one in the Raman spectrum due to the low VC concentration. However, the (baseline-subtracted) number of counts within this band is still 1360 counts per 20 sec integration time, resulting in clear peaks in **Figure 5b**. The signal-to-noise ratio for the vinylene peak is 29, where we have obtained the noise level (~2 counts/sec) from statistical fluctuations in the baseline-subtracted band. The count rate of the vinylene -band increases from 30 counts/sec to 68 counts/sec during the cycle (see **Fig. 5c**). The observed intensity variations in the vinylene band are therefore statistically relevant.

(7) The advancement of the experimental methodology is valuable and can be of importance for future battery research, however the interpretation of results and conclusions require major revision where the authors should either report the data with a higher resolution (at least 2 cm⁻¹) and a proper peak deconvolution; or use a different system to validate their methodology.

> We much appreciate that the Reviewer finds our operando Raman technique valuable and important for future battery research. In response to the Reviewer's suggestion, we have included high-resolution Raman data that reveal the underlying peaks within each of the broader Raman bands in our broadband (but low-resolution) data. In addition, we have also performed (ex-situ) ATR FT-IR

measurements on electrolyte extracted from a cycled cell which provides further insights into, for example, the origin of the vinylene peak (see our response 10 to Reviewer 2, **Fig. R5**).

References (cited by Reviewer 3)

1. Allen, J. L., Borodin, O., Seo, D. M. & Henderson, W. A. Combined quantum chemical/Raman spectroscopic analyses of Li⁺ cation solvation: Cyclic carbonate solvents—Ethylene carbonate and propylene carbonate. *J. Power Sources* **267**, 821–830 (2014).
2. Yang, G. et al. Electrolyte Solvation Structure at Solid–Liquid Interface Probed by Nanogap Surface-Enhanced Raman Spectroscopy. *ACS Nano* **12**, 10159–10170 (2018).
3. Mozhzhukhina, N. et al. Direct Operando Observation of Double Layer Charging and Early Solid Electrolyte Interphase Formation in Li-Ion Battery Electrolytes. *J. Phys. Chem. Lett.* **11**, 4119–4123 (2020).
4. Fortunato, B., Mirone, P. & Fini, G. Infrared and Raman spectra and vibrational assignment of ethylene carbonate. *Spectrochim. Acta Part A Mol. Spectrosc.* **27**, 1917–1927 (1971).

List of changes to references:

References added new manuscript

- [11] Rinkel, B. L. D., Hall, D. S., Temprano, I. & Grey, C. P. Electrolyte Oxidation Pathways in Lithium-Ion Batteries. *J. Am. Chem. Soc.* **142**, 15058–15074 (2020).
(*NMR studies of EC oxidation*)
- [24] Mozhzhukhina, N. et al. Direct Operando Observation of Double Layer Charging and Early Solid Electrolyte Interphase Formation in Li-Ion Battery Electrolytes. *J. Phys. Chem. Lett.* **11**, 4119–4123 (2020). (*as suggested by Reviewer #3, comments 2-3*)
- [25] Seo, D. M. et al. Role of Mixed Solvation and Ion Pairing in the Solution Structure of Lithium Ion Battery Electrolytes. *J. Phys. Chem. C* **119**, 14038–14046 (2015).
(*FT-IR study on carbonate solvents, used to interpret data in new Supplementary Figure 10*)
- [55] Yang, G. et al. Electrolyte Solvation Structure at Solid–Liquid Interface Probed by Nanogap Surface-Enhanced Raman Spectroscopy. *ACS Nano* **12**, 10159–10170 (2018).
(*as suggested by Reviewer #3, comments 2-3*)
- [64] Laszczynski, N., Solchenbach, S., Gasteiger, H. A. & Lucht, B. L. Understanding Electrolyte Decomposition of Graphite/NCM811 Cells at Elevated Operating Voltage. *J. Electrochem. Soc.* **166**, A1853 (2019). (*as suggested by Reviewer #2, comment 11*)
- [65] Ota, H., Sakata, Y., Inoue, A. & Yamaguchi, S. Analysis of Vinylene Carbonate Derived SEI Layers on Graphite Anode. *J. Electrochem. Soc.* **151**, A1659 (2004).
(*cited in response to Reviewer #2, comment 10*)
- [67] Grugeon, S. et al. Towards a better understanding of vinylene carbonate derived SEI-layers by synthesis of reduction compounds. *J. Power Sources* **427**, 77–84 (2019).
(*cited in response to Reviewer #2, comment 10*)

Now redundant references removed from old manuscript:

Benabid, F., Knight, J. C., Antonopoulos, G. & Russell, P. S. J. Stimulated Raman Scattering in Hydrogen-Filled Hollow-Core Photonic Crystal Fiber. *Science* **298**, 399–402 (2002).

Giordano, L. *et al.* Chemical Reactivity Descriptor for the Oxide-Electrolyte Interface in Li-Ion Batteries. *J. Phys. Chem. Lett.* **8**, 3881–3887 (2017).

Giorgini, M. G., Futamatagawa, K., Torii, H., Musso, M. & Cerini, S. Solvation Structure around the Li⁺ Ion in Mixed Cyclic/Linear Carbonate Solutions Unveiled by the Raman Noncoincidence Effect. *J. Phys. Chem. Lett.* **6**, 3296–3302 (2015).

Lim, J. *et al.* Two-Dimensional Infrared Spectroscopy and Molecular Dynamics Simulation Studies of Nonaqueous Lithium Ion Battery Electrolytes. *J. Phys. Chem. B* **123**, 6651–6663 (2019).

Schroder, K. W. *et al.* Effects of Solute–Solvent Hydrogen Bonding on Nonaqueous Electrolyte Structure. *J. Phys. Chem. Lett.* **6**, 2888–2891 (2015).

Tan, C.-Y. & Huang, Y.-X. Dependence of Refractive Index on Concentration and Temperature in Electrolyte Solution, Polar Solution, Nonpolar Solution, and Protein Solution. *J. Chem. Eng. Data* **60**, 2827–2833 (2015).

REVIEWER COMMENTS

Reviewer #2 (Remarks to the Author):

The authors have addressed all of the original comments. This improves the quality of the submitted manuscript.

Please be careful to remove any internal comments between the authors, as on page 15 line 256.

Reviewer #3 (Remarks to the Author):

My main concern with the reported methodology was that the spectral resolution of reported Raman spectra is very low in order to draw any conclusions related to the Li⁺ solvation dynamics. I appreciate that authors have performed the high-resolution ex situ spectra and have included them in the manuscript. However, this only partially addresses my concern since the reported operando data is still comprised of a low-resolution data. With a resolution of 27 cm⁻¹ it is not possible to resolve individual Raman peaks, which means that the bulk solvent and coordinated solvent bands cannot be resolved. Therefore, it is not possible to distinguish whether the results represent the change in Li-ion solvation, or the bulk solvent molecular ratio, so it is not possible to draw any certain conclusions about Li-ion solvation.

In the rebuttal letter authors remark that present manuscript presents a "proof-of-principle study" and "our method opens up great opportunities for future experiments on Li⁺ solvation dynamics using high-resolution spectra such as the one presented in Fig. R12b. However, we feel these fall outside the scope of the current proof-of-principle demonstration of operando Raman spectroscopy on full-cell batteries."

I agree that manuscript indeed presents interesting methodology and can be published as a "proof-of-principle demonstration". However, authors should edit manuscript accordingly and remove/rephrase the conclusions about Li-ion solvation dynamics that are derived from low resolution spectra. Instead, it should be acknowledged (and proved) in the manuscript that the reported methodology has a potential to monitor solvation dynamics if performed operando with high resolution, but the presented data constitutes only a proof-of-concept.

The following corrections should be performed:

- Please comment on the potential to use the methodology for operando high-resolution experiments. What is the measurement time compared to low-resolution experiments? Is time still short enough for performing operando measurements?
- How much of a change can be expected in Li-ion solvation during the battery cycling: what percentage of electrolyte is expected to be decomposed and how would it translate into the bulk electrolyte ratio of the components? How much intensity change of Raman bands can be expected?
- lines 22-24 "Our data reveal.... changes in the lithium-ion solvation dynamics". Should be rephrased to indicate there is a potential to monitor the solvation dynamics.
- line 204 "Monitoring Li⁺ solvation" and line 210 "The complex interplay of lithium solvation dynamics with solvents (EC, EMC)" It would be more appropriate to indicate that both solvents coordinated to Li, as well as bulk solvents molecules are monitored, which cannot be distinguished with low-resolution set-up.

- lines 296-298 "Li⁺ solvation: The increase of the EC C=O relative to the EMC C=O Raman mode measured in regions II-IV (Fig. 5d) is likely related to Li⁺ solvation. The shape of the spectra seems to indicate preferential Li⁺ solvation at the EC C=O compared to EMC C=O " The observed Raman bands increase could be both related to bulk molecules ratio or solvation. The conclusion about preferential solvation is therefore not supported by presented data.

- Lines 341-342 "The change in intensity of the EC breathing mode (880-915 cm⁻¹) may be linked to a modification in solvation structure of Li⁺- O=C of EC" Authors should include additional explanations.

- line 361 "We have observed significant changes in lithium solvation with carbonate solvents " and lines 366-367 "Our results demonstrate that hollow-core fibre spectroscopy allows detailed studies of how (electro)chemical degradation of electrolytes affects lithium solvation." Conclusions should be rephrased.

- line 256 needs a correction of "#####add Didi's paper here "

We are delighted by the positive response to our letter by Reviewer #2 (“addressed all of the original comments”, “improves the quality of the submitted manuscript”) and are highly encouraged by the positive comments from Reviewer #3 (“interesting methodology”, “can be published as a proof-of-principle demonstration”).

Reviewer #2:

The authors have addressed all of the original comments. This improves the quality of the submitted manuscript.

Please be careful to remove any internal comments between the authors, as on page 15 line 256.

> We thank the reviewer for this positive feedback and for all their comments that were instrumental in improving the overall quality of our work. We have edited the text and removed the internal comment.

Reviewer #3:

1) My main concern with the reported methodology was that the spectral resolution of reported Raman spectra is very low in order to draw any conclusions related to the Li⁺ solvation dynamics. I appreciate that authors have performed the high-resolution ex situ spectra and have included them in the manuscript. However, this only partially addresses my concern since the reported operando data is still comprised of a low-resolution data. With a resolution of 27 cm⁻¹ it is not possible to resolve individual Raman peaks, which means that the bulk solvent and coordinated solvent bands cannot be resolved. Therefore, it is not possible to distinguish whether the results represent the change in Li-ion solvation, or the bulk solvent molecular ratio, so it is not possible to draw any certain conclusions about Li-ion solvation.

In the rebuttal letter authors remark that present manuscript presents a “proof-of-principle study” and “our method opens up great opportunities for future experiments on Li⁺ solvation dynamics using high-resolution spectra such as the one presented in Fig. R12b. However, we feel these fall outside the scope of the current proof-of-principle demonstration of operando Raman spectroscopy on full-cell batteries.”

I agree that manuscript indeed presents interesting methodology and can be published as a “proof-of-principle demonstration”. However, authors should edit manuscript accordingly and remove/rephrase the conclusions about Li-ion solvation dynamics that are derived from low resolution spectra. Instead, it should be acknowledged (and proved) in the manuscript that the reported methodology has a potential to monitor solvation dynamics if performed operando with high resolution, but the presented data constitutes only a proof-of-concept.

> We thank the reviewer for suggesting to include this clarification about the scope of this work, and its possibilities. We agree that in reporting this proof-of-concept of a unique methodology we could rephrase our conclusion about Li-ion solvation dynamics. We now revised the text to better convey this point and removed stronger claims on effects of Li-ion solvation on Raman bands. However we believe evidence is also accessible even from lower resolution spectra (which conversely give wide enough spectral range to capture a broader range of information simultaneously) using more detailed data analysis (such as standard deconvolution and/or Principal Components Analysis, although out of the scope of this report). We thus provide careful clarification.

The following corrections should be performed:

1) Please comment on the potential to use the methodology for operando high-resolution experiments. What is the measurement time compared to low-resolution experiments? Is time still short enough for performing operando measurements?

> This is indeed an important question to consider, for integration times required in the high-resolution experiments in Supplementary Figure 7. The integration time for these is the same (20 seconds) as for the spectra acquired with the low-resolution/higher-efficiency grating. Even though

the grating efficiency is reduced from ~85% to ~50% (Fig. R1), this does not substantially limit the overall collection efficiency. Considering that typical formation/diagnostic battery cycling is performed at low C-rates (C/20), it is clear that even 60s integration times (compared to 20s used) do not hinder the use of this methodology for operando studies. A drawback of the high-resolution gratings (1200 grooves/mm) is the inability to track different Raman bands simultaneously (limited spectral window).

Figure R1: Comparison of the efficiency of the gratings in our experiments. (Top) Low-resolution (300 g/mm) grating: a 1000 nm blaze grating (orange curve) with an efficiency of ~85% at 845 nm was used. (Bottom) High-resolution (1200 g/mm) grating: a 500 nm blaze grating (orange curve) with an efficiency of ~50% at 845 nm was used. Curves obtained from:

<https://www.princetoninstruments.com/learn/calculators/grating-dispersion>.

How much of a change can be expected in Li-ion solvation during the battery cycling: what percentage of electrolyte is expected to be decomposed and how would it translate into the bulk electrolyte ratio of the components? How much intensity change of Raman bands can be expected?

> This is an important point. According to ex-situ data shown in Supplementary Figure 8, we expect 10% intensity changes for a 0.1 M change in Lithium concentration. In addition to this effect, we note that local Li-ion concentrations can be affected by the presence of Faradaic currents, which can significantly change the Li-ion concentration in the sample electrolyte.

On the other hand, the EC decomposition from SEI formation is expected to be negligible compared to its initial volume in the first cycle, with significant concentration changes limited to the SEI thickness (~100nm max, <https://iopscience.iop.org/article/10.1149/2.1101902jes>). However, this effect can indeed become significant in the subsequent cycles where we observe the formation of linear carbonates (vinylene) species and, therefore, a modification of the structure of the C=O bands (Supplementary Figure 10). We now explicitly note this in the text.

3) lines 22-24 “Our data reveal... changes in the lithium-ion solvation dynamics”. Should be rephrased to indicate that there is a potential to monitor the solvation dynamics.

> We agree and changed the text accordingly: “Our data reveal changes in the ratio of carbonate solvents and electrolyte additives as a function of the cell voltage and **show the potential to track the lithium-ion solvation dynamics.**”

4) line 204 “Monitoring Li⁺ solvation ...” and line 210 “The complex interplay of lithium solvation dynamics with solvents (EC, EMC)” It would be more appropriate to indicate that both solvents coordinated to Li, as well as bulk solvent molecules are monitored, which cannot be distinguished with low-resolution set-up.

> Thanks for raising this point. We modified the caption of Fig. 4 as follows: “The deconvolution based on the solid purple line in **a suggests that the peaks comprise a combination of solvent C=O stretch bands for un-solvated Li⁺ (dark blue, ~1750 cm⁻¹, for EMC and dark red, ~1805 cm⁻¹, for EC) and solvated Li⁺ (light blue, 1735 cm⁻¹, for Li⁺EMC and dark red, ~1787 cm⁻¹, for Li⁺EC). **With the broadband (and thus lower-resolution) grating used here, differences between solvated and non-solvated Li⁺ cannot be simply resolved.**”**

5) lines 296-298 “Li⁺ solvation: The increase of the EC C=O relative to the EMC C=O Raman mode measured in regions II-IV (Fig. 5d) is likely related to Li⁺ solvation. The shape of the spectra seems to indicate preferential Li⁺ solvation at the EC C=O compared to EMC C=O ” The observed Raman bands increase could be both related to bulk molecules ratio or solvation. The conclusion about preferential solvation is therefore not supported by presented data.

> Thank you for this helpful comment. We now updated the text to clarify this: “Li⁺ solvation: The increase of EC C=O relative to the EMC C=O Raman mode measured in regions II-IV (Fig. 4d) **could be related to Li⁺ concentration changes and/or Li⁺ solvation of the solvent molecules,⁶⁶ which is consistent with the ex-situ data in Supplementary Figure 8. However, the increase could also be caused by changes in the bulk EC/EMC ratio, which cannot be resolved at the spectral resolution used in Fig. 4.**”

6) Lines 341-342 “The change in intensity of the EC breathing mode (880-915 cm⁻¹) may be linked to a modification in solvation structure of Li⁺-O=C of EC” Authors should include additional explanations.

> Indeed additional detail would make this paragraph clearer and we now expanded our explanation by explicitly referring to the solvated and non-solvated EC breathing modes and how they are affected by Li⁺ coordination: “The EC breathing band at 880-915 cm⁻¹ comprises two distinct modes at 893 cm⁻¹ (not solvated) and 903 cm⁻¹ (solvated), which are resolved in the high-resolution spectra in Supplementary Figure 7. The intensity ratio between these two modes is known to be particularly sensitive to Li-ion coordination.^{24,63}”

7) line 361 “We have observed significant changes in lithium solvation with carbonate solvents ” and lines 366-367 “Our results demonstrate that hollow-core fibre spectroscopy allows detailed studies of how (electro)chemical degradation of electrolytes affects lithium solvation.” Conclusions should be rephrased.

> We have rephrased this part of the conclusions to reflect the Reviewer’s earlier suggestions and to avoid making premature conclusions: “Our results **show the potential of hollow-core fibre spectroscopy to study** how (electro)chemical degradation of electrolytes affects lithium solvation. **Future work employing the narrower-band high-resolution spectrometer configuration, will clearly track the solvated and non-solvated Raman modes of various targeted electrolyte components.**”

8) line 256 needs a correction of “#####add Didi’s paper here ”

> Noted with thanks. We have removed this internal comment.

REVIEWERS' COMMENTS

Reviewer #3 (Remarks to the Author):

Authors have addressed all the previous concerns which improved the manuscript quality. I therefore recommend the manuscript to be accepted at Nature communications in its current form. Please also note that ex situ and in situ should be written without hyphen.